# Yerba mate (*Ilex paraguariensis*) genome provides new insights into convergent evolution of caffeine biosynthesis

Federico A Vignale[1]*[†], Andrea Hernandez Garcia[2][†], Carlos P Modenutti[3,4], Ezequiel J Sosa[3,4], Lucas A Defelipe[1], Renato Oliveira[5], Gisele L Nunes[5], Raúl M Acevedo[6], German F Burguener[7], Sebastian M Rossi[8], Pedro D Zapata[8], Dardo A Marti[9], Pedro Sansberro[6], Guilherme Oliveira[5], Emily M Catania[10], Madeline N Smith[10], Nicole M Dubs[10], Satish Nair[11,12], Todd J Barkman[10]*, Adrian G Turjanski[3,4]*

[1]European Molecular Biology Laboratory - Hamburg Unit, Hamburg, Germany; [2]Department of Biochemistry, University of Illinois at Urbana-Champaign, Urbana, United States; [3]IQUIBICEN-CONICET, Ciudad Universitaria, Pabellón 2, Ciudad Autonoma de Buenos Aires, Argentina; [4]Departamento de Química Biológica, Facultad de Ciencias Exactas y Naturales, Universidad de Buenos Aires, Ciudad Universitaria, Pabellón 2, Ciudad Autónoma de Buenos Aires, Argentina; [5]Instituto Tecnológico Vale, Belém, Brazil; [6]Laboratorio de Biotecnología Aplicada y Genómica Funcional, Instituto de Botánica del Nordeste (IBONE-CONICET), Facultad de Ciencias Agrarias, Universidad Nacional del Nordeste, Corrientes, Argentina; [7]Department of Plant Sciences, University of California, Davis, Davis, United States; [8]Instituto de Biotecnología de Misiones, Facultad de Ciencias Exactas, Químicas y Naturales, Universidad Nacional de Misiones (INBIOMIS-FCEQyN-UNaM), Misiones, Argentina; [9]Instituto de Biología Subtropical, Universidad Nacional de Misiones (IBS-UNaM-CONICET), Posadas, Argentina; [10]Department of Biological Sciences, Western Michigan University, Kalamazoo, United States; [11]Carl R. Woese Institute for Genomic Biology, University of Illinois at Urbana-Champaign, Urbana, United States; [12]Center for Biophysics and Quantitative Biology, University of Illinois at Urbana Champaign, Urbana, United States

*For correspondence:
federico.vignale@embl-hamburg. de (FAV);
todd.barkman@wmich.edu (TJB);
aturjans@gmail.com (AGT)

[†]These authors contributed equally to this work

**Competing interest:** The authors declare that no competing interests exist.

**Abstract** Yerba mate (YM, *Ilex paraguariensis*) is an economically important crop marketed for the elaboration of mate, the third-most widely consumed caffeine-containing infusion worldwide. Here, we report the first genome assembly of this species, which has a total length of 1.06 Gb and contains 53,390 protein-coding genes. Comparative analyses revealed that the large YM genome size is partly due to a whole-genome duplication (Ip-α) during the early evolutionary history of *Ilex*, in addition to the hexaploidization event (γ) shared by core eudicots. Characterization of the genome allowed us to clone the genes encoding methyltransferase enzymes that catalyse multiple reactions required for caffeine production. To our surprise, this species has converged upon a different biochemical pathway compared to that of coffee and tea. In order to gain insight into the structural basis for the convergent enzyme activities, we obtained a crystal structure for the terminal enzyme in the pathway that forms caffeine. The structure reveals that convergent solutions have evolved for substrate positioning because different amino acid residues facilitate a different substrate orientation such that efficient methylation occurs in the independently evolved enzymes in YM and coffee. While our results show phylogenomic constraint limits the genes coopted for

convergence of caffeine biosynthesis, the X-ray diffraction data suggest structural constraints are minimal for the convergent evolution of individual reactions.

## Editor's evaluation

This very important study combines genomics, biochemistry, structural biology and ancestral sequence reconstruction to address the basis of caffeine biosynthesis in Yerba mate, a species that is phylogenetically unrelated to other plants, namely coffee and tea, in which this pathway has been studied before. The manuscript reports the first draft genome sequence for yerba mate and provides convincing evidence for the identity and characteristics of enzymes for caffeine biosynthesis. The authors are able to propose structural constraints for the convergent evolution of individual reactions. The work will be of interest to plant and evolutionary biologists and anyone studying natural product biosynthesis.

## Introduction

In the genomic era, hundreds of plant species have had their nucleotide sequences determined to provide unprecedented insight into the genetic basis of many traits. One of the few species of economic importance for which no genomic data exist is *Ilex paraguariensis* var. *paraguariensis* A. St. Hilaire (Aquifoliaceae), colloquially known as yerba mate (YM), which is a caffeinated diploid tree-species ($2n = 2x = 40$) endemic to the subtropical rainforests of South America (*Niklas, 1987*). The dried leaves and twigs of this dioecious evergreen are used to prepare a traditional infusion named mate, or chimarrão, widely consumed around the world. Approximately 300,000 ha are cultivated with this tree crop, with Argentina responsible for 80% of worldwide production (*Heck and de Mejia, 2007*). The mate infusion has been shown to have numerous beneficial effects in humans including as an antioxidant (*Sánchez Boado et al., 2015*; *Gugliucci, 1996*; *Vieira et al., 2010*), antidiabetic (*Kang et al., 2012*; *Ríos et al., 2015*), as well as central nervous system stimulant (*Santos et al., 2015*), among others. Several bioactive compounds have been identified in YM that might be responsible for its effects, including terpenes, flavonoids, phenolics, and methylxanthines (*Heck and de Mejia, 2007*). Although its stimulant properties are best known and mostly related to caffeine content, little is known about the genetic and biochemical mechanisms of how YM synthesizes this, or any, of its important metabolites. Despite the recent release of three other *Ilex* genome sequences (*Kong et al., 2022*; *Xu et al., 2022*; *Yao et al., 2022*), none of the species produce caffeine, making the genetic basis for convergent evolution of this trait in YM unclear.

Convergent evolution has occurred throughout the tree of life and is particularly rampant in plants (*Sackton and Clark, 2019*) where examples of repeated origins of morphological (*Thorogood et al., 2018*), anatomical (*Wan et al., 2018*), physiological (*Yang et al., 2017*), and biochemical (*Pichersky and Lewinsohn, 2011*) traits have been documented. Caffeine (CF) is a xanthine alkaloid that has independently evolved no less than six times across angiosperms and has implications for pollination, insect defence, and allelopathy (*Anaya et al., 2006*; *Stevenson et al., 2017*). There are multiple biosynthetic routes to caffeine possible within the xanthine alkaloid network (*Figure 1*). Within the Rosid genera *Theobroma*, *Paullinia*, and *Citrus*, sequential methylation of xanthine (X), 1- and/or 3-methylxanthine (1X, 3X) and, finally, either theophylline (TP) or theobromine (TB) leads to caffeine (*Huang et al., 2016*). In contrast, the Asterids, *Coffea* and *Camellia*, appear to sequentially methylate xanthosine (XR), 7-methylxanthine (7X), and theobromine (TB) to yield caffeine (*Ashihara et al., 1996*; *Suzuki and Takahashi, 1976*; *Figure 1*). Regardless of which pathway is utilized, species differ in terms of which SABATH enzyme family members were convergently recruited to synthesize caffeine: xanthine methyltransferase (XMT) is used by *Citrus* and *Coffea* while the paralogous caffeine synthase (CS) is used by *Camellia*, *Theobroma*, and *Paullinia* (*Huang et al., 2016*; *Kato et al., 1996*; *Uefuji et al., 2003*; *Figure 1*). Convergence appears to also extend to the mutational level, since different amino acid replacements to homologous regions of CS and XMT enzymes appear to govern the evolution of substrate preference switches (*O'Donnell et al., 2021*). However, it remains unclear whether mutations lead to convergent three-dimensional protein structures to confer convergent substrate interactions and catalysis by the enzymes. Because XMT- and CS-type enzymes have been convergently recruited in both Rosids and Asterids to catalyse the same or different pathways, it suggests

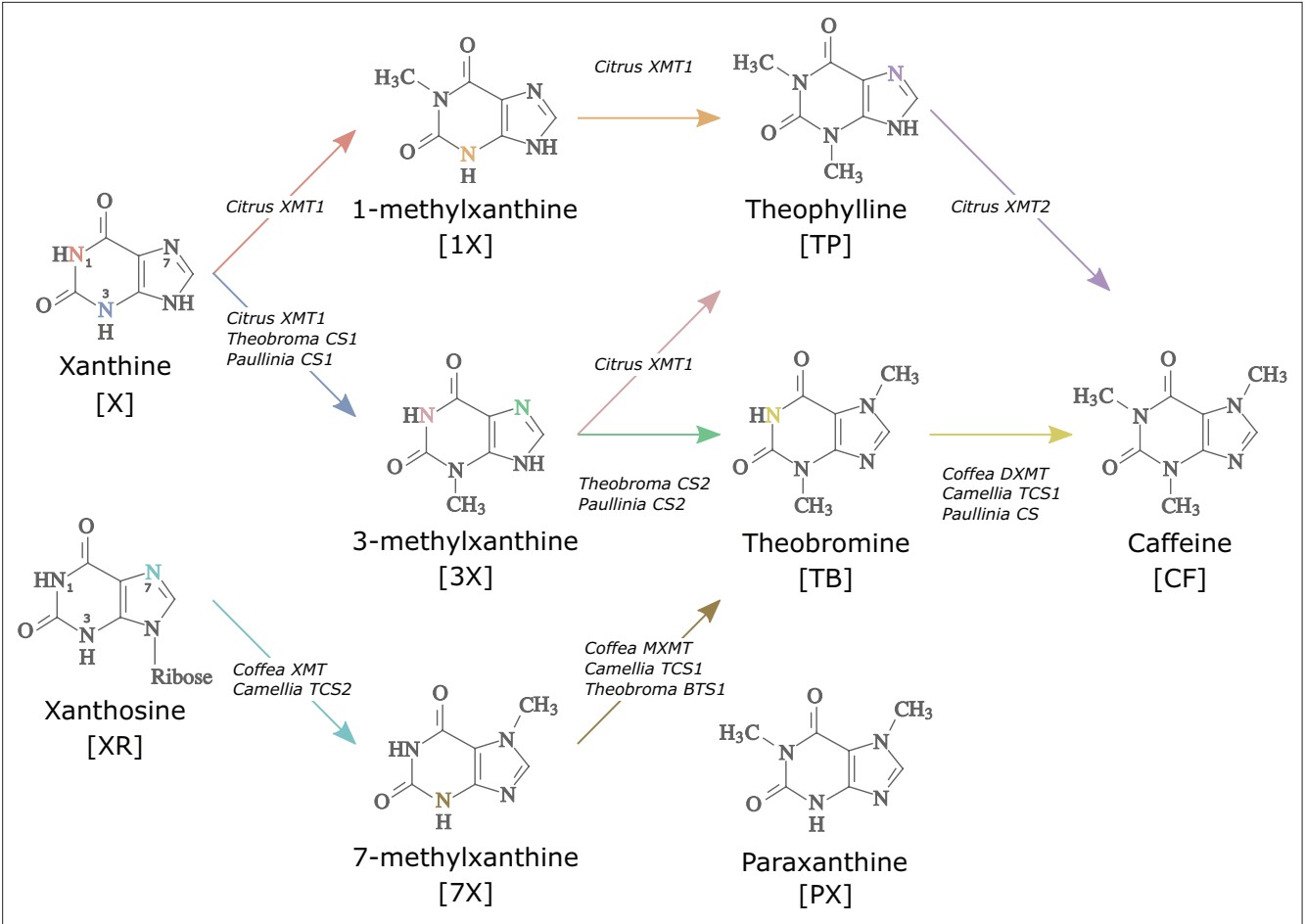

**Figure 1.** Biosynthetic routes to caffeine within the xanthine alkaloid network. CF, caffeine; PX, paraxanthine; TB, theobromine; TP, theophylline; 1X, 1-methylxanthine; 3X, 3-methylxanthine; 7X, 7-methylxanthine; XR, xanthosine; X, xanthine. Nitrogen atoms are coloured to match the arrows corresponding to the enzymes that methylate them. Adapted with permission from *O'Donnell et al., 2021*.

considerable evolutionary lability underlying caffeine production in plants (*Huang et al., 2016*). As a result, it is difficult to predict what sets of genes, biochemical reactions, and structural properties might lead to the evolution of caffeine biosynthesis in YM, or any plant, a priori.

Although some transcriptomic resources have been generated for YM (*Acevedo et al., 2019*; *Debat et al., 2014*; *Fay et al., 2018*), a complete genome sequence has the potential to advance our understanding of the metabolic potential of this important crop and facilitate improvement. Here, we describe the first draft genome of YM and report on its composition, organization, and evolution. The genomic sequence enabled us to uncover the genetic, biochemical, mutational, and structural bases for convergent evolution of caffeine in YM. Our comparative analyses of caffeine-producing enzymes across angiosperms reveal how convergence may be the result of constrained evolutionary genomic potential but relatively unconstrained structural potential.

## Results and discussion
### YM genome sequencing, assembly, and annotation

The YM genome was sequenced combining Illumina and PacBio sequencing technologies. With Illumina sequencing, we generated ~263.2 Gb of short reads from various DNA fragment sizes (350 bp, 550 bp, 3 kbp, 8 kbp, and 12 kbp), while with PacBio sequencing, we generated ~77.5 Gb of long reads. These reads represent ~158.5- and ~49.3-fold base-pair coverage of the genome, respectively (*Table 1*). The total assembly length was ~1.06 Gb and consisted of 10,611 scaffolds (≥1 kb) with an N50 length of ~510.8 kb (*Table 2*). To assess the completeness of the genome, we aligned

**Table 1.** Statistics of the genome sequencing data of yerba mate.

| Library | Number of reads | Read length | Total length | Coverage |
|---|---|---|---|---|
| Pair-end 350 bp #1 | 360,653,408 | 101 | 36.4 Gbp | 21.8× |
| Pair-end 350 bp #2 | 368,746,464 | 101 | 37.2 Gbp | 22.3× |
| Pair-end 550 bp | 356,261,246 | 101 | 36 Gbp | 21.5× |
| Mate-pair 3 kbp #1 | 415,398,586 | 101 | 30.3 Gbp | 18.2× |
| Mate-pair 3 kbp #2 | 410,588,934 | 101 | 30 Gbp | 17.9× |
| Mate-pair 3 kbp #3 | 343,059,350 | 101 | 25 Gbp | 15× |
| Mate-pair 8 kbp | 393,202,256 | 101 | 34.6 Gbp | 20.7× |
| Mate-pair 12 kbp | 415,478,776 | 101 | 33.7 Gbp | 20.1× |
| PacBio long reads | 19,514,627 | 50 bp to 61 kbp | 77.5 Gbp | 49.3× |
| Total | | | 341 Gbp | 207.8× |

the available YM transcriptome reads (*Acevedo et al., 2019*; *Debat et al., 2014*; *Fay et al., 2018*) and the YM genomic short reads generated in this study with the assembly: 99.3% of the former and 99.5% of the latter were mapped. The GC content of the genome assembly was 36.33% (*Table 2*), similar to that of other eudicots (33.70–38.20 GC%) (*Singh et al., 2016*) and almost identical to that of *Ilex polyneura* (36.08 GC%) (*Yao et al., 2022*), *Ilex asprella* (36.25 GC%) (*Kong et al., 2022*), and *Ilex latifolia* (36.44 GC%) (*Xu et al., 2022*), the only three *Ilex* species with sequenced genomes. About 64.63% of the genome assembly was composed of repetitive sequences, of which ~36.22% were retrotransposons, ~1.80% were DNA transposons, ~0.74% were simple repeats, and ~0.15% were low complexity regions. Long terminal-repeat retrotransposons of the Gypsy and Copia families were the most abundant transposable elements, as observed in many sequenced plant genomes (*Galindo-González et al., 2017*), followed by long interspersed nuclear elements (LINEs) and hobo-Activator transposons, among others (*Table 3*).

A total of 53,390 protein-coding genes were predicted in the genome, with a mean coding sequence length of 3062 bp and 4.23 exons per gene. Of these, 41,483 (~77.63%) could be annotated

**Table 2.** Statistics of the genome assembly of yerba mate.

| Metric | Value |
|---|---|
| # scaffolds (≥1000 bp) | 10,611 |
| # scaffolds (≥5000 bp) | 9343 |
| # scaffolds (≥10,000 bp) | 8951 |
| # scaffolds (≥25,000 bp) | 5944 |
| # scaffolds (≥50,000 bp) | 2595 |
| Total length (≥50,000 bp) | 887,124,725 |
| # scaffolds | 10,611 |
| Largest scaffold | 7,402,063 |
| Total length | 1,064,802,823 |
| GC (%) | 36.33 |
| N50 | 510,878 |
| N75 | 132,523 |
| L50 | 506 |
| L75 | 1461 |
| # N's per 100 kbp | 1976.99 |

**Table 3.** Classification and distribution of repetitive DNA elements in yerba mate.

| | Number | Length occupied (bp) | Percentage of the genome (%) |
|---|---|---|---|
| Class I retrotransposons | 421,599 | 385,714,532 | 36.22 |
| SINEs | 840 | 154,298 | 0.01 |
| Penelope | 0 | 0 | 0.00 |
| LINEs | 35,433 | 17,109,207 | 1.61 |
| CRE/SLACS | 0 | 0 | 0.00 |
| L2/CR1/Rex | 575 | 135,549 | 0.01 |
| R1/LOA/Jockey | 443 | 76,937 | 0.01 |
| R2/R4/NeSL | 0 | 0 | 0.00 |
| RTE/Bov-B | 8599 | 2,126,765 | 0.20 |
| L1/CIN4 | 25,816 | 14,769,956 | 1.39 |
| LTR retrotransposons | 385,326 | 368,451,027 | 34.60 |
| BEL/Pao | 709 | 266,632 | 0.03 |
| Ty1/Copia | 98,237 | 67,631,136 | 6.35 |
| Gypsy/DIRS1 | 216,472 | 274,526,515 | 25.78 |
| Retroviral | 0 | 0 | 0.00 |
| Class II DNA transposons | 45,427 | 19,116,209 | 1.80 |
| hobo-Activator | 21,335 | 6,378,850 | 0.60 |
| Tc1-IS630-Pogo | 0 | 0 | 0.00 |
| En-Spm | 0 | 0 | 0.00 |
| MuDR-IS905 | 0 | 0 | 0.00 |
| PiggyBac | 0 | 0 | 0.00 |
| Tourist/Harbinger | 5870 | 2,846,548 | 0.27 |
| Others | 0 | 0 | 0.00 |
| Unclassified | 990,080 | 269,430,122 | 25.30 |
| Total interspersed repeats | 674,260 | 863 | 63.32 |
| Small RNA | 4362 | 718,762 | 0.07 |
| Satellites | 0 | 0 | 0.00 |
| Simple repeats | 185,507 | 7,911,080 | 0.74 |
| Low complexity | 31,856 | 1,606,255 | 0.15 |

with GO terms, EC numbers or Pfam domains. In addition, we identified 4530 non-coding RNA genes, including 2670 small nucleolar RNAs, 815 transfer RNAs, 471 ribosomal RNAs, 348 small nuclear RNAs, and 226 micro RNAs (Appendix 1, *Appendix 1—tables 1–3*). To further assess the completeness of the assembly, we aligned the scaffolds with the KOG (*Tatusov et al., 2003*) and DEG (*Luo et al., 2014*) databases, determining that 98% of the core gene families from the KOG database and 97.5% of the *Arabidopsis thaliana* DEG subset were present. Then, we performed a Benchmarking Universal Single-Copy Orthologs (BUSCO) (*Manni et al., 2021*) assessment using the eudicot ODB10 database. Among 2326 conserved single-copy genes, ~96.20% were retrieved, of which ~78.80% were complete and single copies, ~17.40% were complete and in duplicates, ~3.10% were fragmented, and only ~0.70% were missing. These results suggest that the coding region of the assembly is nearly complete. The number of estimated genes for YM is higher than the ca. 39,000 reported from the genome sequences of other *Ilex* species (*Kong et al., 2022*; *Xu et al., 2022*; *Yao et al., 2022*).

This could be at least partly due to the larger genome size of YM as estimated from flow cytometry relative to the other species (*Gottlieb and Poggio, 2015*).

## Evolutionary analysis of YM genome provides evidence of whole-genome duplication in an early *Ilex* ancestor

Most plant lineages have experienced ancient polyploidization events followed by massive duplicate gene losses and genome rearrangements, which may have contributed to the evolution of developmental and metabolic complexity (*Landis et al., 2018*; *Sankoff and Zheng, 2018*). Recent transcriptome-based analyses (*One Thousand Plant Transcriptomes Initiative, 2019*; *Zhang et al., 2020b*) reported an ancient polyploidization event in the *Ilex* lineage around 60 Ma (Cretaceous–Paleogene boundary), based on phylogenomic and synonymous substitution rate ($K_s$) evidence. Evolutionary analyses of *I. polyneura* (*Yao et al., 2022*) and *I. latifolia* (*Xu et al., 2022*) genomes also provided evidence of a shared *Ilex*-specific whole-genome duplication (WGD). As YM is the first American holly to have its genome sequenced, we performed synteny-based analyses of its genome to deepen our understanding of Aquifoliales evolution (*Figure 2*, *Figure 2—figure supplement 1*). The $K_s$ distribution of YM paralogues (*Figure 2B*) revealed a significant peak with a median $K_s$ value of ~0.37, not shared with the rest of the eudicot genomes analysed (*Figure 2B*, *Figure 2—figure supplement 1*). This confirms the lineage-specific polyploidization event (Ip-α) previously reported in *Ilex* (*One Thousand Plant Transcriptomes Initiative, 2019*; *Xu et al., 2022*; *Yao et al., 2022*; *Zhang et al., 2020b*), in addition to the shared ancestral WGT-γ which is indicated by a median $K_s$ value of ~1.4 (*Figure 2B*). A WGD in the common ancestor of *Ilex* species is further supported by 2:1 syntenic depth ratios between the YM genome and the coffee and grape genomes, which did not experience additional duplication events after the ancestral WGT-γ (*Figure 2C*). In order to determine the age of Ip-α, we used two different phylogenies (*Figure 2A*). The plastid genome phylogeny supports the monophyly of Aquifoliales as the first diverging clade of campanulids (*Magallón et al., 2015*); the alternative nuclear genome phylogeny supports *Ilex* in Aquifoliales I as an early branching lineage of lamiids (*Zhang et al., 2020b*). With the former phylogeny, we estimated the age of the WGD event between 48.75 and 69.63 Ma while, with the latter, divergence was estimated at 49.43 and 70.62 Ma (*Figure 2A*). Both estimates are consistent with that of *Zhang et al., 2020a* and validate the age of Ip-α near the origin of *Ilex*, which is estimated between 43 and 89 Ma (*Yao et al., 2021*).

## Convergent evolution of caffeine biosynthesis in YM

In order to determine the genes and biochemical pathway responsible for caffeine biosynthesis in YM, we used bioinformatic analyses to identify SABATH enzyme family members in the genome (*Huang et al., 2016*; *Kato et al., 1996*; *Uefuji et al., 2003*). There appear to be 28 full-length SABATH genes in YM that encode members of the functionally diverse clades of the family, including SAMT (*Ross et al., 1999*) and JMT (*Seo et al., 2001*), among others (*Figure 3A*). Our phylogenetic analysis showed that although the YM genome does not appear to encode XMT-type caffeine-producing enzymes like *Coffea* and *Citrus*, it does contain three recently and tandemly duplicated genes that encode CS-type enzymes, IpCS1, IpCS2, and IpCS3 (*Figure 3A, C*, Appendix 2). The duplicated IpCS1–3 are 86–91% identical at the amino acid level and are expressed at highest levels in caffeine-accumulating tissue (*Figure 3B*). IpCS1–3 also appear to be of recent origin, since non-caffeine accumulating *Ilex* species only have a single gene or gene fragment in the syntenic region (*Figure 3C*). In *Camellia*, *Theobroma*, and *Paullinia*, recent duplications of the CS-type enzymes responsible for the successive steps of xanthine alkaloid methylation have also independently occurred (*Figure 3A*; *Huang et al., 2016*; *O'Donnell et al., 2021*). Two other YM genes encode IpCS4 and 5, but these are not syntenic with IpCS1–3 and are not highly expressed in any tissues studied (*Figure 3B*); therefore, we did not characterize them further.

To investigate the biochemical activities of the enzymes encoded by the three CS-type genes, we cloned them into bacterial expression vectors and determined heterologous protein functions. One enzyme, IpCS1, appears to primarily methylate X to catalyse the formation of 3X (*Figure 4*). A second enzyme, IpCS2, shows activity only with 3X to produce TB, while a third enzyme, IpCS3, exhibits a preference to methylate TB to form CF (*Figure 4*). Thus, collectively, these three enzymes appear capable of catalysing a complete pathway from xanthine to caffeine. The apparent $K_M$ for the preferred substrates of all three enzymes ranges from 85 to 197 µM, and the $k_{cat}/K_M$ estimates are

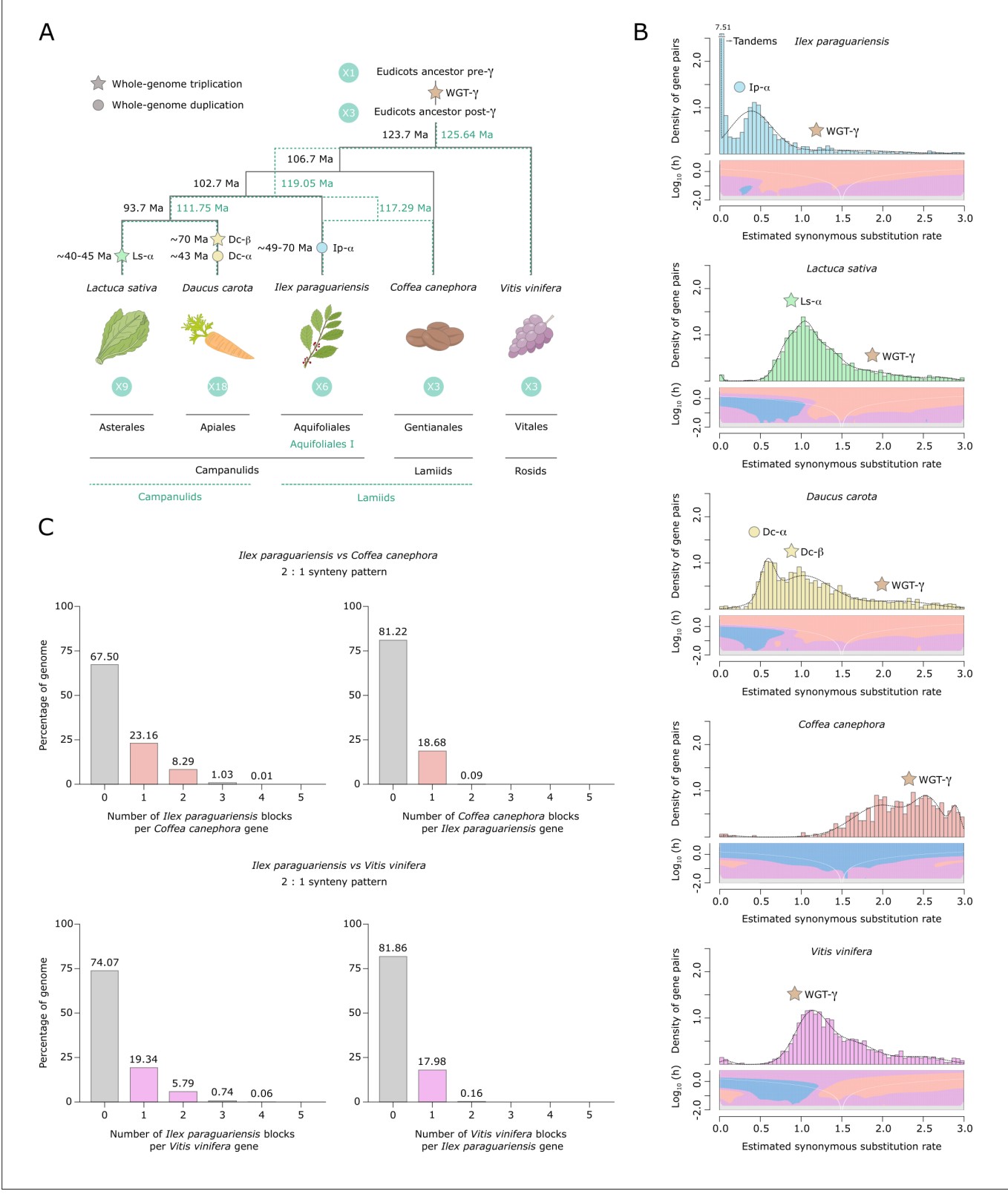

**Figure 2.** Yerba mate genome duplication history. (**A**) Evolutionary scenario of the eudicot genomes of *Lactuca sativa*, *Daucus carota*, *Ilex paraguariensis*, *Coffea canephora*, and *Vitis vinifera*, from their ancestor pre-γ. The plastid genome phylogeny is represented with solid black lines, while the multiple nuclear genome phylogeny is represented with green dashed lines. Paleopolyploidizations are shown with coloured dots (duplications) and stars (triplications). Divergence time estimates for the lineages, as well as age estimates for the *L. sativa* and *D. carota* paleopolyploidizations

*Figure 2 continued on next page*

*Figure 2 continued*

were obtained from the literature (*Iorizzo et al., 2016*; *Magallón et al., 2015*; *Reyes-Chin-Wo et al., 2017*; *Zhang et al., 2020b*). Ma, million years ago. (**B**) $K_s$ distributions with Gaussian mixture model and SiZer analyses of *I. paraguariensis* (blue), *L. sativa* (green), *D. carota* (yellow), *C. canephora* (red), and *V. vinifera* (purple) paralogues. SiZer maps below histograms identify significant peaks at corresponding $K_s$ values. Blue represents significant increases in slope, red indicates significant decreases, purple represents no significant slope change, and grey indicates not enough data for the test. (**C**) Comparative genomic synteny analyses of *I. paraguariensis* with *C. canephora* and *V. vinifera*.

The online version of this article includes the following figure supplement(s) for figure 2:

**Figure supplement 1.** $K_s$ distributions with Gaussian mixture model and SiZer analyses of *I. paraguariensis* and *L. sativa* (green), *D. carota* (yellow), *C. canephora* (red), and *V. vinifera* (purple) orthologues.

comparable to those determined for other caffeine biosynthetic enzymes (*O'Donnell et al., 2021*; *Table 4*, *Figure 4—figure supplement 1*). Further evidence for this biosynthetic pathway has been reported by $^{14}$C xanthine tracer studies in young leaf segments of *I. paraguariensis* that showed radio-activity in 3X and TB in addition to CF (*Yin et al., 2015*). A pathway from X→3X→TB→CF has also been reported for *Theobroma* and *Paullinia* using CS-type SABATH enzymes (*Huang et al., 2016*). Like *Huang et al., 2016*, this represents another departure from the long-assumed pathway to caffeine biosynthesis (XR→7X→TB→CF) as reported in coffee and tea (*Figure 1*). This instance in *Ilex* is particularly notable since YM is an Asterid, like coffee and tea. The fact that *Ilex*, *Theobroma*, and *Paullinia* convergently recruited CS genes that independently duplicated and evolved to encode enzymes with similar substrate preferences to catalyse a common pathway to caffeine, in spite of their divergence more than 100 Ma (*Yang et al., 2020*), is remarkable and suggests a high degree of genetic constraint governing the repeated origin of this trait.

While the substrate preferences shown in *Figure 4* suggest pathway flux from X→3X→TB→CF, IpCS1 also shows secondary activity with 7X to produce TB and IpCS3 can catalyse the formation of CF from paraxanthine (PX) (*Figure 4A*). Thus, flux through other branches of the xanthine alkaloid biosynthetic network (*Figure 1*) cannot be excluded. However, it is not clear how 7X or PX would be produced in planta since none of the three enzymes studied here is capable of their formation; therefore, these secondary activities may not be physiologically relevant. In addition, it has been proposed that TP may also be a precursor to caffeine biosynthesis in *I. paraguariensis* based on radioisotopic feeding studies (*Yin et al., 2015*), although its levels in plant tissues are 30–160 times lower than TB (*Negrin et al., 2019*). Our in vitro enzyme assays provide no experimental evidence for that biosynthetic route; however, it is possible that additional MT enzymes from the SABATH (or other) gene family not characterized in this study may perform such reactions. Alternatively, if the exogenously supplied TP was first catabolized to 3X in YM tissues, then the caffeine detected previously (*Yin et al., 2015*) could have been synthesized via the route described above for IpCS2 + IpCS3 (*Figure 4*).

## The caffeine biosynthetic pathway in YM evolved from ancestral networks with different inferred flux

Caffeine is produced within only one small lineage of *Ilex* that diverged and experienced CS gene duplication (*Figure 3*) within the last 11 million years (*Negrin et al., 2019*; *Yao et al., 2021*) which indicates that the pathway has only recently evolved. The nature by which novel multistep biochemical pathways evolve is a central question in biology (*Noda-Garcia et al., 2018*). To investigate the caffeine pathway origin in YM, we used Ancestral Sequence Reconstruction (*Dean and Thornton, 2007*; *Thornton, 2004*) to study AncIpCS1 and AncIpCS2, the ancestors of the three modern-day enzymes implicated in caffeine biosynthesis in YM (*Figure 5*, *Figure 5—figure supplements 1–4*). The ancestral enzyme, AncIpCS1, which gave rise to all three modern-day YM enzymes, exhibits highest relative activity with X, 3X, and 7X (*Figure 5A*). Methylation of 7X by AncIpCS1 occurred at the N3 position resulting in TB synthesis, whereas xanthine methylation occurred at either the N1 or N3 position to form 1X and 3X, respectively (*Figure 5B*, *Figure 5—figure supplement 5A*). AncIpCS1 was capable of methylation of 3X at N1 to produce TP, while methylation at the N7 position led to TB formation (*Figure 5B*, *Figure 5—figure supplement 5A*). These data demonstrate that, although AncIpCS1 could not produce caffeine, it could methylate xanthine alkaloids at 3 different positions of the planar heterocyclic ring structures and this combination of activities would have allowed for the ancestor of YM to produce a cocktail of 1X, 3X, TP, and TB by flux through multiple branches of the xanthine alkaloid biosynthetic network with a single enzyme (*Figure 5B*).

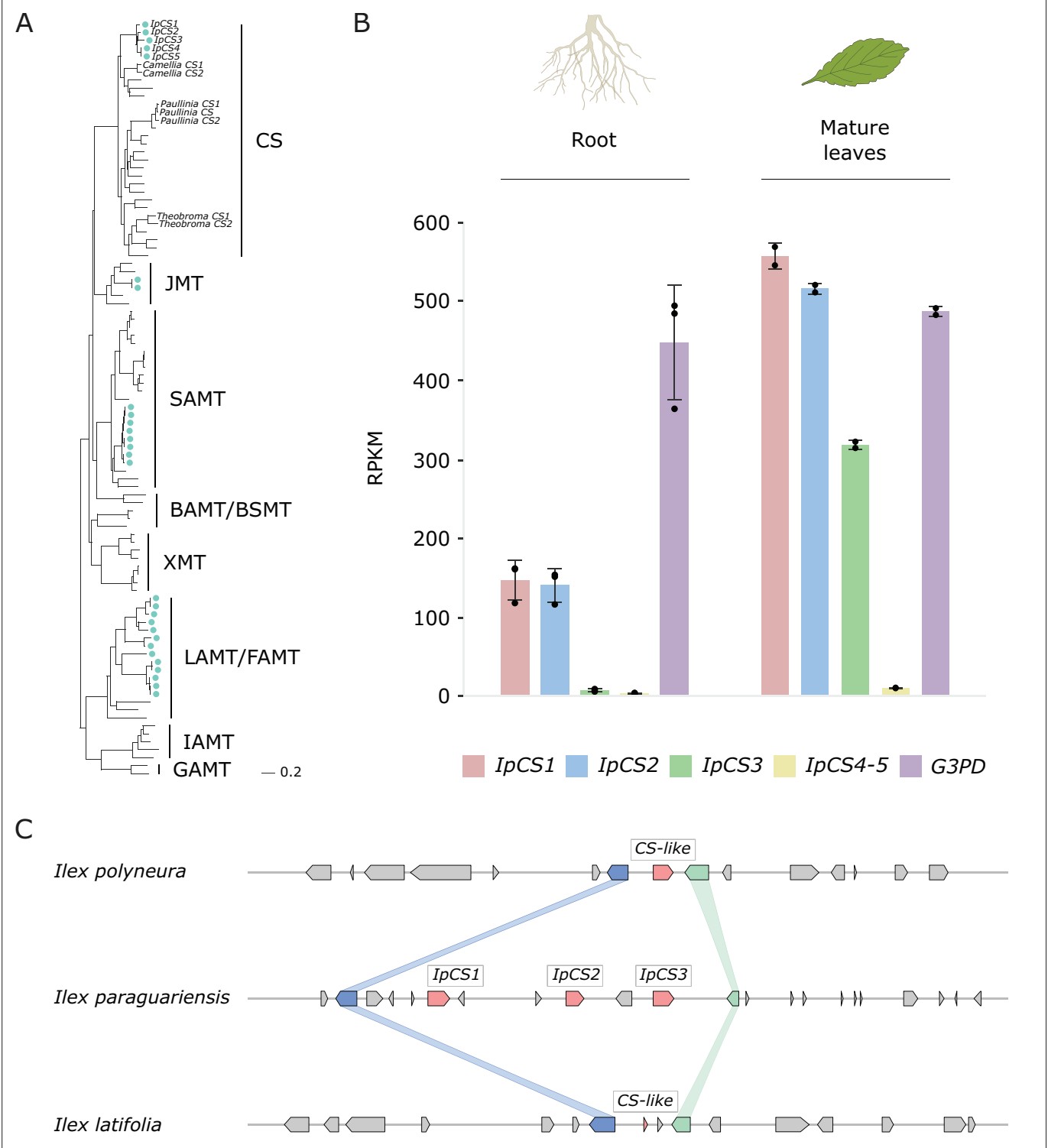

**Figure 3.** The yerba mate (YM) genome encodes three recently duplicated CS-type SABATH proteins that are expressed in caffeine-producing tissues. (**A**) SABATH gene tree estimate (LnL = −34,265.473) shows the placement of full-length YM proteins (marked by blue-green dots) within clades that have published functions. GAMT, gibberellin MT; IAMT, indole-3-acetic acid MT; LAMT/FAMT, loganic/farnesoic acid MT; BAMT/BSMT, benzoic/salicylic acid MT; XMT, xanthine alkaloid MT used for caffeine biosynthesis in *Coffea* and *Citrus*; SAMT, salicylic acid MT; JMT, jasmonic acid MT; CS, caffeine synthase in *Theobroma*, *Camellia,* and *Paullinia*. Accession numbers for all sequences are provided in *Figure 3—source data 1*. (**B**) Gene expression analysis of IpCS1–5 in root (*n* = 3) and mature leaves (*n* = 2) as indicated by the relative abundance of YM transcriptome reads mapped to the IpCS1–5 transcripts.

*Figure 3 continued on next page*

*Figure 3 continued*

RPKM, reads per kilobase per million mapped reads. Error bars indicate standard deviation from the mean. Housekeeping gene: *G3PD*, glyceraldehyde-3-phosphate dehydrogenase. (**C**) Synteny-based analysis of the CS genomic region for *I. paraguariensis*, *I. polyneura*, and *I. latifolia*.

The online version of this article includes the following source data for figure 3:

**Source data 1.** Accession numbers of SABATH sequences used for phylogenetic analysis in *Figure 3*.

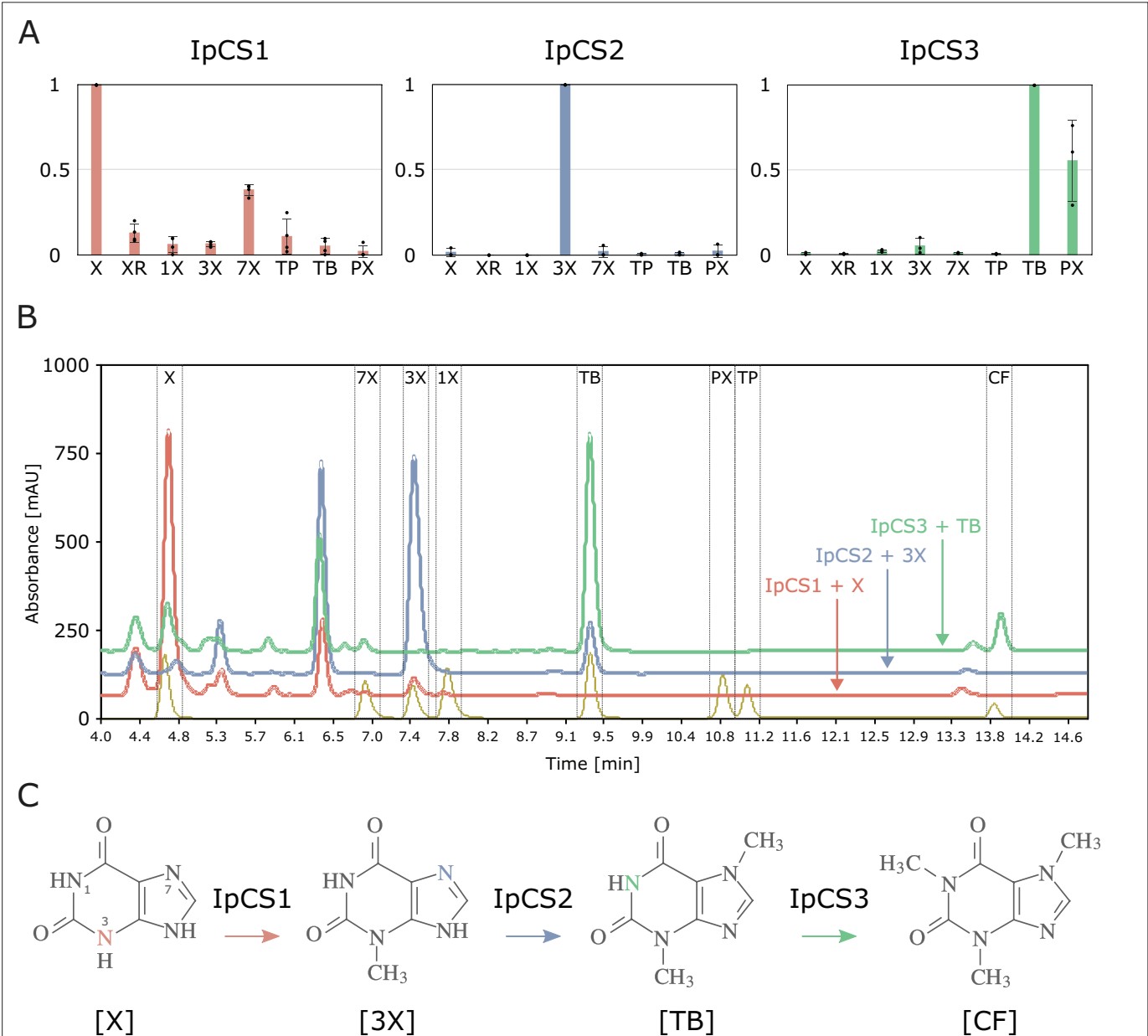

**Figure 4.** SABATH enzymes have evolved to catalyse the biosynthesis of caffeine in yerba mate. (**A**) Relative enzyme activitiy of IpCS1 (*n* = 4), IpCS2 (*n* = 3), and IpCS3 (*n* = 3) SABATH enzymes with eight xanthine alkaloid substrates. (**B**) High-performance liquid chromatography (HPLC) traces showing products formed by three encoded caffeine synthase (CS)-type enzymes. Absorbance at 254 nm is shown. (**C**) Proposed biosynthetic pathway for caffeine in yerba mate. X, xanthine; XR, xanthosine; 1X, 1-methylxanthine; 3X, 3-methylxanthine; 7X, 7-methylxanthine; TP, theophylline; TB, theobromine; PX, paraxanthine. Coloured atoms and arrows indicate atoms that act as methyl acceptors for a given reaction.

The online version of this article includes the following figure supplement(s) for figure 4:

**Figure supplement 1.** Michaelis–Menten curves used to estimate kinetic parameters for (**A**) IpCS1, (**B**) IpCS2, and (**C**) IpCS3 (*n* = 2).

**Table 4.** Apparent enzyme kinetic parameter estimates for yerba mate caffeine biosynthetic enzymes with selected substrates.

| Enzyme (substrate) | $K_M$ (μM) | $k_{cat}$ (1/s) | $k_{cat}/K_M$ (s$^{-1}$ M$^{-1}$) |
|---|---|---|---|
| IpCS1 (X) | 85.05 | 0.0009 | 10.11 |
| IpCS2 (3X) | 197.08 | 0.0031 | 15.77 |
| IpCS3 (TB) | 151.19 | 0.0029 | 19.36 |

After gene duplication of AncIpCS1, one daughter enzyme ultimately gave rise to IpCS1, which exhibits preference to methylate xanthine to produce 3X (*Figure 5A*). The other daughter enzyme, AncIpCS2, appears to have maintained highest activity with X, 3X, and 7X like AncIpCS1 (*Figure 5A*). However, unlike its ancestor, AncIpCS1, AncIpCS2 evolved high relative activity with 7X to produce not just TB, but also PX by methylation at the N1 position (*Figure 5A*, *Figure 5—figure supplement 5B*). AncIpCS2 retained the ancestral activity of AncIpCS1 with xanthine to produce 1X, but also evolved the ability to methylate X at the N7 position (*Figure 5B*, *Figure 5—figure supplement 5B*). This enzyme also retained ancestral activity with 3X to produce only TB by N7 methylation but lost the ability to methylate 3X at the N1 position to form TP. These activities of AncIpCS2 would have allowed for ancestral flux to produce 1X, 7X, TB, and PX but not caffeine. Because a YM ancestor could have possessed both AncIpCS2 and a descendant of AncIpCS1, AncIpCS1′ (*Figure 5B*), additional pathway flux is possible. If AncIpCS1′ retained activities of its ancestor, AncIpCS1, then the ancestral *Ilex* species could have also produced 3X and TP making for an even more diverse array of xanthine alkaloids in its tissues (*Figure 5B*). It has been shown that the xanthine alkaloids, 1X, 3X, and TP, can bind to modern-day rat adenosine receptors (*Daly et al., 1983*). Therefore, if these molecules were to accumulate in ancestral *Ilex* tissues, they could have conferred a selective advantage which would likely result in retention of the ancient genes. Ultimately, once gene duplication led to the generation of the three modern-day CS-type enzymes in YM, pathway flux could be channelled away from intermediates like 1X and TP such that the modern-day pathway to caffeine evolved (*Figure 5B*). Not only did the modern-day CS enzymes of YM evolve to catalyse a pathway from X>3X>TB>CF from ancestral biosynthetic networks of different products, *Theobroma* and *Paullinia* also independently evolved enzymes with similar properties (*Huang et al., 2016*). And, they did so from ancestral pathways that, like YM, had alternative ancestral fluxes (*O'Donnell et al., 2021*). While it could be due to chance alone that all three lineages converged to catalyse a similar pathway from differing ancestral networks, it is also possible that it was advantageous to specialize for flux to TB via X and 3X because either it is more enzymatically favourable or these intermediates have greater adaptive value than other structural isomers.

## Protein crystal structure of IpCS3 reveals convergent structural basis for methylation of theobromine to form caffeine

We successfully crystallized and determined the 2.7 Å resolution structure of IpCS3 (PDB ID: 8UZD), that converts TB into CF. This enzyme crystallizes as a holo-homodimer, bound to both of its reaction products: *S*-adenosyl-homocysteine (SAH) and caffeine (*Figure 6A*, *Table 5*). As is typical for the SABATH family of methyltransferases, IpCS3 exhibits a Rossman-like fold composed of seven β-strands surrounded by five α-helices which bind the methyl-donor *S*-adenosyl-L-methionine (SAM), as well as an α-helical cap which binds the methyl-acceptor theobromine (*McCarthy and McCarthy, 2007*; *Petronikolou et al., 2018*; *Zhao et al., 2008*; *Zubieta et al., 2003*). This structural information of the enzyme bound to both of its products, SAH and caffeine, facilitates an in-depth comparison of the active site structures of the caffeine-producing CS-type enzyme found in *Ilex* to the XMT-type enzyme in *Coffea canephora* (*McCarthy and McCarthy, 2007*) (CcDXMT) to determine the extent to which convergence of physicochemical properties of the active site has allowed for independent specialization for theobromine methylation by the paralogous SABATH enzymes. Although the IpCS3 structure was obtained in complex with its product, caffeine (*Figure 6—figure supplement 1*), it can be assumed that the binding mode is conserved for its precursor, theobromine. Indeed, our computational modelling of theobromine in the active site of IpCS3 predicts it to be oriented as we have discerned from the diffraction data (*Figure 6—figure supplement 2*). Thus, in the following

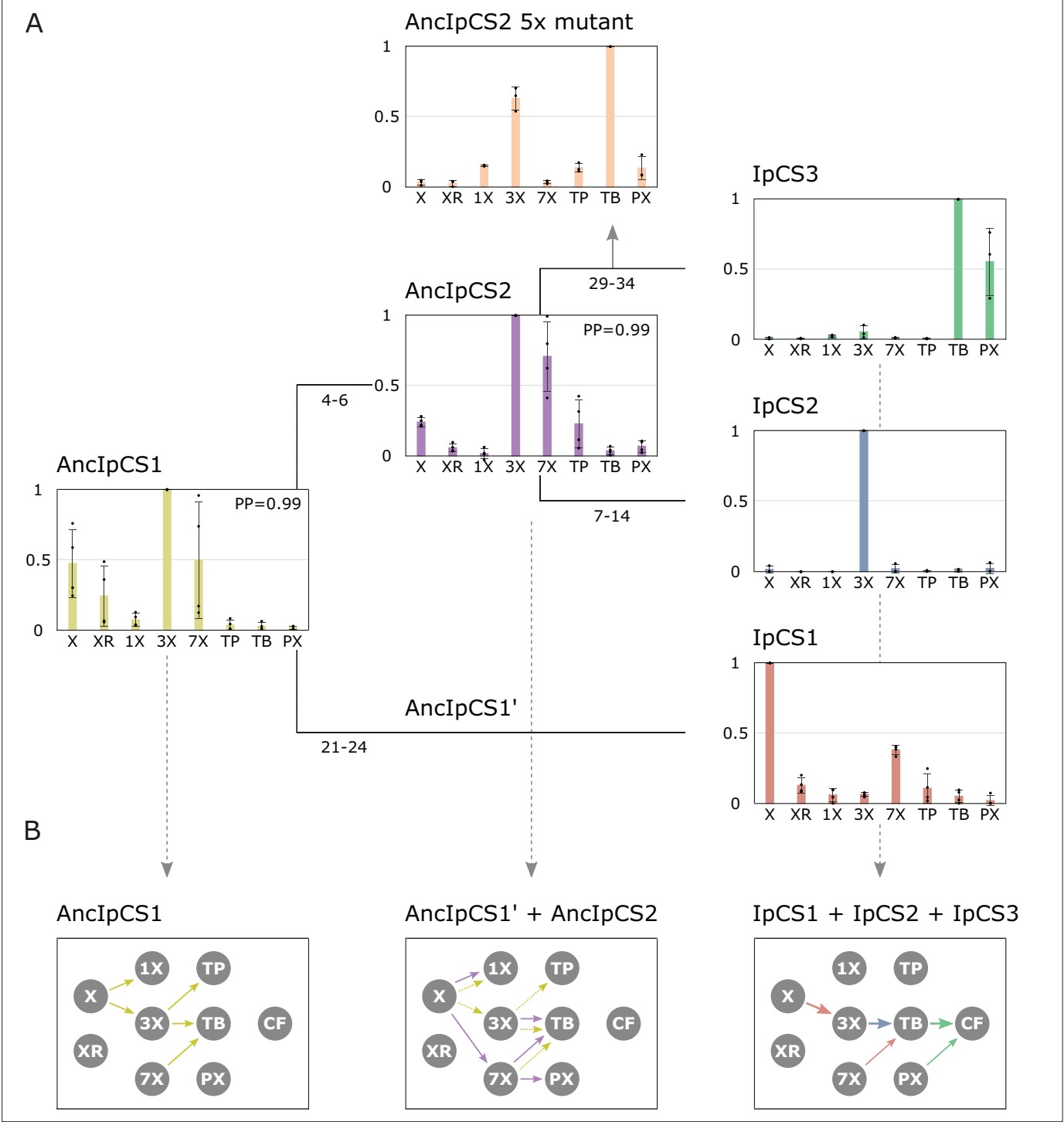

**Figure 5.** Ancestral sequence resurrection reveals ancestral xanthine alkaloid pathway flux. (**A**) Simplified evolutionary history of three yerba mate (YM) xanthine alkaloid-methylating enzymes and their two ancestors, AncIpCS1 and AncIPCS2. Average site-specific posterior probabilities (PP) for each ancestral enzyme estimate are provided. Numbers below each branch of the phylogeny represents the number of amino acid replacements between each enzyme shown. These two ancestral relative activity charts (*n* = 4) show the averaged activities of two allelic variants of each enzyme. Relative substrate preference is also shown for the AncIPCS2 mutant enzyme (*n* = 3) in which five amino acid positions, A22G, R23C, T25S, H221N, and Y265C, that are inferred to have been replaced during the evolution of IpCS3, were changed. (**B**) Inferred pathway flux is shown for the antecedent pathways that could have been catalysed by the ancestral or modern-day combinations of enzymes that would have existed at three time points in the history of the enzyme lineage. Arrows linking metabolites are coloured according to the activities detected from each enzyme shown in panel A. Dotted arrows are shown for AncIpCS1' because it is unknown what characteristics it would possess; it is assumed that it would have at least catalysed the formation

*Figure 5 continued on next page*

*Figure 5 continued*

of 3X from X since both its ancestor and descendant enzyme do so. X, xanthine; XR, xanthosine; 1X, 1-methylxanthine; 3X, 3-methylxanthine; 7X, 7-methylxanthine; TP, theophylline; TB, theobromine; PX, paraxanthine.

The online version of this article includes the following figure supplement(s) for figure 5:

**Figure supplement 1.** SABATH enzyme family phylogenetic tree used for obtaining ancestral sequence estimates for the clade including IpCS1–3 of Aquifoliales (log-likelihood = −46,631.672).

**Figure supplement 2.** Caffeine synthase enzyme family phylogenetic tree used for obtaining alternative ancestral sequence estimates for AncIpCS1 and 2 (log-likelihood = −7032.8928).

**Figure supplement 3.** Alignment of the two estimated amino acid sequences for AncIpCS1 that were biochemically characterized in *Figure 5*.

**Figure supplement 4.** Alignment of the two estimated amino acid sequences for AncIpCS2 that were biochemically characterized in *Figure 5*.

**Figure supplement 5.** High-performance liquid chromatography (HPLC) traces for xanthine alkaloid products formed by ancestral *Ilex* caffeine synthase (CS) enzymes.

---

comparisons, the atomic numbering for the theobromine precursor will be used to facilitate comparison to the CcDXMT structure.

In both CcDXMT and IpCS3, there are several conserved residues, shared by nearly all SABATH enzymes (*Figure 6—figure supplements 3 and 4*), that form the active site pocket and appear to play important roles in binding many different substrates (*Petronikolou et al., 2018*; *Zubieta et al., 2003*). His160 and Trp161 in CcDXMT are in the same relative positions as His155 and Trp156 in IpCS3 (*Figure 6B, C*). These residues are ca. 3 Å from TB and participate in H-bonding but to different atoms of the substrate. In CcDXMT, the NE2 of His160 and NE1 of Trp161 form hydrogen bonds to carbonyl O2 of TB when positioned for N1 methylation; yet, in the structure of IpCS3 these corresponding side chain groups form hydrogen bonds to O6 of TB. Despite these two residues being conserved for H-bonding, the substrates are rotated 180° along an axis going through N1 and C4. Thus, the conserved His and Trp residues interact with opposing carbonyls in TB/CF but still position the substrate for N1 methylation (*Figure 6B, C*).

On the other hand, there are residues that differ between the two enzymes but appear to provide for important substrate interactions. Specifically, in the structure of CcDXMT, the hydroxyl group of Ser237 allows specific hydrogen bonding with N9 to position TB for N1 methylation (*Figure 6C*). In IpCS3, His236 is found at the homologous position in the structure. Nevertheless, its involvement in H-bonding with N9 is uncertain as the distance between nitrogen atoms is ca. 4 Å. Tyr368 of CcDXMT is found to participate in π–π interactions with the ring structure of TB *Lanzarotti et al., 2020*; yet in IpCS3, Asn353 is found in the homologous position and the amine forms a hydrogen bond with N9 due to its proximity within 3.2 Å, which is additionally stabilized by Asn221 and His236 (*Figure 6B*). The caffeine-producing CS-type enzymes found in *Camellia sinensis* (CsTCS1) and *Paullinia cupana* (PcCS), may share the same interaction pattern observed with Asn353 in IpCS3 because the homologous Thr in CsTCS1 or Gln in PcCS could potentially form a hydrogen bond with N9 (*Figure 6B*, *Figure 6—figure supplements 3 and 4*). Because the residues in these positions of IpCS3 and CcDXMT differ yet contribute to TB binding, these independent replacements represent convergent structural solutions for N1 methylation of the substrate.

In order to experimentally test for the functional importance of the active site residues identified in the crystal structure of IpCS3 for the evolution of TB methylation preference, we performed site-directed mutagenesis. We chose to mutate five amino acid positions that appear to be important for governing xanthine alkaloid methylation in IpCS3 and other CS-type enzymes (*Jin et al., 2016*; *O'Donnell et al., 2021*; *Wang et al., 2023*; *Yoneyama et al., 2006*); these included A22G, R23C, T25S, H221N, and Y265C (*Figure 6—figure supplements 3 and 4*). When we mutated all five amino acid residues simultaneously in AncIpCS2, we found that activity with TB increased dramatically relative to 3X and all other xanthine alkaloid substrates (*Figure 5A*). Thus, these five sites appear to be crucial for the evolution of TB methylation in the history of the YM lineage and further indicate that convergence of caffeine biosynthesis in different species is a result of amino acid replacements at these sites. The homologous sites to H221N and Y265C in Theacrine synthase from *Camellia assamica* were also shown by mutagenesis to be important for the evolution of trimethyluric acid methylation (*Zhang et al., 2020a*) thereby providing further support for the functional significance of these positions for xanthine alkaloid binding.

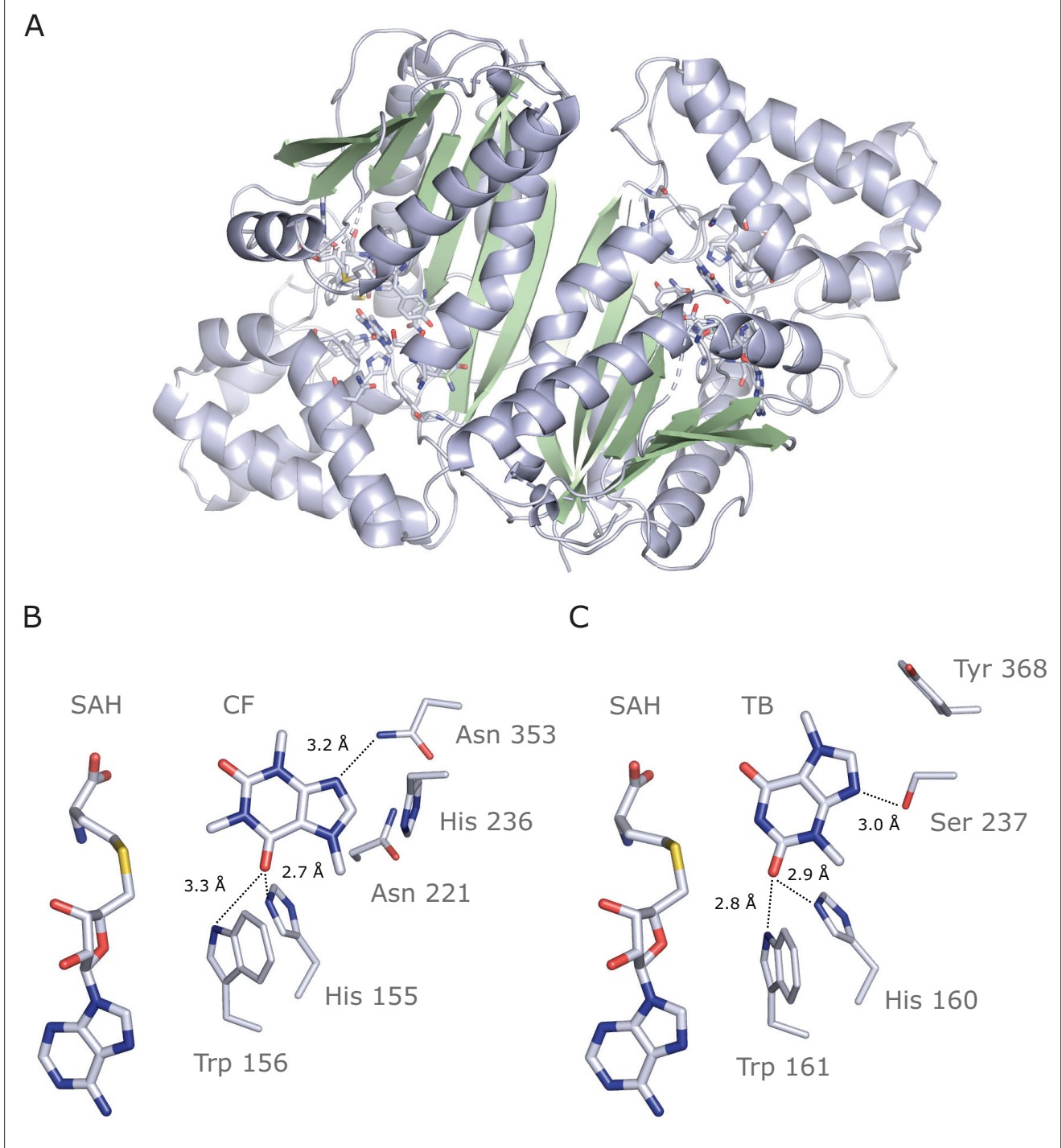

**Figure 6.** Crystal structure of IpCS3 in complex with caffeine (CF) and *S*-adenosyl-homocysteine (SAH) and comparison with the active site of *Coffea canephora* DXMT. (**A**) Overview of the crystal structure of IpCS3 (PDB ID: 8UZD) depicting the active site of the enzyme in complex with CF and SAH. (**B**) Relevant residues in IpCS3 for ligand recognition. (**C**) Relevant residues in CcDXMT (PDB ID: 2EFJ) for ligand recognition. Protein residues are displayed as lines with carbon atoms coloured in bluewhite while small molecules – CF, theobromine (TB), and SAH – are drawn as sticks. Colour code for the rest of the atoms: nitrogen (blue), oxygen (red), and sulphur (yellow). Hydrogen bond interactions are indicated as black dotted lines.

The online version of this article includes the following figure supplement(s) for figure 6:

**Figure supplement 1.** Crystal structure of IpCS3 displaying a difference Fourier map (Fo − Fc) contoured to 2.0 σ (blue) showing bound SAH and CF.

**Figure supplement 2.** Theobromine and caffeine are oriented the same way in the active site of IpCS3.

*Figure 6 continued on next page*

*Figure 6 continued*

**Figure supplement 3.** Comparative amino acid alignment of xanthine methyltransferase (XMT) and caffeine synthase (CS) sequences (1–209 of IpCS1) shows convergent changes predicted to participate in substrate binding and promote methylation preference switches.

**Figure supplement 4.** Comparative amino acid alignment of xanthine methyltransferase (XMT) and caffeine synthase (CS) sequences (210–365 of IpCS1) shows convergent changes predicted to participate in substrate binding and promote methylation preference switches.

## Computational modelling of IpCS1 and IpCS2 active sites predict convergent substrate-binding residues for xanthine and 3-methylxanthine methylation

Previous studies used site-directed mutagenesis of two sequence regions in CS-type caffeine biosynthetic enzymes from *Theobroma* (TcCS1/2) and *Paullinia* (PcCS1/2) to uncover the mutational basis for the convergent evolution of substrate preference switches towards their preferred substrates, X and 3X (*O'Donnell et al., 2021*). In order to determine whether the same regions were convergently mutated in IpCS1 and IpCS2, and provide binding interactions with X and 3X, respectively, AlphaFold2 (*Mirdita et al., 2022*) models and subsequent docking studies were performed (*Figure 7*, *Figure 7—figure supplement 1*). Modelling of substrate binding in the predicted active sites of IpCS1 and IpCS2 (*Figure 7A and B*) shows that the preferred substrates have optimal binding orientations that would result in methylation to form the products that were experimentally detected in our assays shown in *Figure 4*. From our docking simulations, IpCS1 residues W156, N221, and Y265 are positioned for hydrogen bonding with the carbonyl moieties of xanthine to position N3 for methyl transfer from SAM (*Figure 7A*). Although *Theobroma* and *Paullinia* CS1 enzymes, as well as *Citrus* XMT1, specialized for xanthine methylation also possess W156 and Y265 in the homologous positions (*Figure 6—figure supplements 3 and 4*), these residues are highly conserved among nearly all angiosperm SABATH enzymes. On the other hand, the homologous position to N221 which is important for IpCS1 did not change concomitantly with the evolution of X preference in either *Theobroma* or *Paullinia* (*Figure 6—figure supplements 3 and 4*); instead, when *Theobroma* and *Paullinia* 'region III' was mutated, activity with X improved (*O'Donnell et al., 2021*). Because IpCS1 was not mutated in the homologous region III, there appear to be convergent solutions allowing for efficient positioning of X for 3X biosynthesis among these enzymes. In the case of IpCS2, two hydrogen bond donors, S24 and T25, appear to contribute to the positioning of 3X in the active site (*Figure 7B*). This homologous region was experimentally mutated in *Theobroma* and *Paullinia* CS2 enzymes and improved specialization for 3X methylation in both, although the actual substitutions are different in each case (*O'Donnell et al., 2021*). Thus, this may represent an additional instance where convergent mutations of the same region lead to specialization for 3X methylation. If crystal structures could be generated for all of these caffeine-producing enzymes in the future, even more detailed insights about active site architecture could be gleaned and would further enhance our understanding of these convergent activities.

## Comparative phylogenomic analyses of caffeine biosynthetic genes reveal historical constraints to convergent gene co-option

Many nearly ubiquitous specialized metabolites involved in defence, development and floral scent are produced by SABATH enzyme family members that appear to be conserved across diverse angiosperm lineages, such as SAMT that methylates salicylic acid (*Dubs et al., 2022*) and IAMT that methylates indole-3-acetic acid (*Zhao et al., 2008*). However, caffeine is sporadically distributed among disparate angiosperm lineages and seems to have only recently evolved by convergence in a few distantly related orders (*Huang et al., 2016*). Our comparative evolutionary genomic analysis of the CS and XMT syntenic regions across angiosperm (*Figure 8*) indicates that predicting which SABATH locus a given lineage might co-opt for caffeine biosynthesis is more dependent upon the idiosyncratic history of gene loss than phylogenetic relatedness. For example, in the case of the CS syntenic region used for caffeine biosynthesis in YM and *Theobroma*, *Coffea* lacks a CS orthologue and none can be detected from its genome (*Figure 8—figure supplement 1A*). Thus, only XMT was historically available for recruitment in *Coffea*. Conversely, YM appears to have lost any vestiges of XMT orthologues known to be responsible for caffeine biosynthesis in *Coffea* and *Citrus* (*Figure 8—figure supplement 1B–D*). This lack of genomic potential may be seen as an evolutionary constraint to gene recruitment for caffeine biosynthesis in *Coffea* and YM.

**Table 5.** Data collection and refinement statistics of IpCS3 structure bound to *S*-adenosyl-homocysteine (SAH) and caffeine.

**IpCS3 in complex with SAH and caffeine**

| | |
|---|---|
| PDB | 8UZD |
| **Data collection** | |
| Wavelength (Å) | 0.9786 |
| Resolution (Å) | 2.72 |
| Resolution range[a*] | 37.00–2.72 |
| | (2.82–2.72) |
| Space group | $P\,4_1\,2_1\,2$ |
| Cell dimensions | |
| *a, b, c* (Å) | 82.67, 82.67, 226.09 |
| *α, β, γ* (°) | 90.00, 90.00, 90.00 |
| Total reflections | 43,818 |
| Unique reflections | 21,910 |
| Multiplicity[a*] | 2.0 (2.0) |
| Completeness (%)[a*] | 99.89 (100.00) |
| $<I/\sigma I>$[a] | 25.79 (2.87) |
| $R_{merge}$[a,b†*] (%) | 0.0223 (0.2168) |
| $R_{meas}$ (%)[a*] | 0.0315 (0.3066) |
| $CC_{1/2}$[a*] | 0.999 (0.878) |
| **Refinement** | |
| Resolution (Å) | 2.72 |
| No. reflections | 21,909 |
| $R_{work}$[c‡]/$R_{free}$[d§] | 0.194/0.248 |
| No. atoms | |
| Protein | 5,216 |
| CFF + SAH | 80 |
| Water | 48 |
| *B*-factors | |
| Protein | 63.38 |
| CFF + SAH | 84.48 |
| Water | 48.19 |
| Bond lengths (Å) | 0.004 |
| Bond angles (°) | 1.112 |

*[a]Numbers in parentheses refer to the highest resolution shell.

†[b]$R_{merge} = \Sigma |I_i - <I_i>|/\Sigma I_{ir}$, where $I_i$ = the intensity of the *i*th reflection and $<I_i>$ = mean intensity.

‡[c]$R_{work} = \Sigma |F_o - F_c|/\Sigma |F_o|$, where $F_o$ and $F_c$ are the observed and calculated structure factors, respectively.

§[d]$R_{free}$ was calculated as for $R_{work}$, but on a test set comprising 5% of the data excluded from refinement.

A broader phylogenetic perspective on the XMT and CS syntenic regions provides further insight into genomic canalization and allows for predictions about the underlying genetic basis for caffeine biosynthesis in as-of-yet characterized lineages. As shown in *Figure 8*, several angiosperm lineages have neither XMT nor CS and this may explain why caffeine has apparently never evolved in the large and diverse orders Brassicales, Asterales, Solanales and Lamiales even though it has been shown to be advantageous in transgenic plants (*Kim et al., 2011*; *Kim et al., 2006*). In the case of *Cola*, a caffeine-producing genus from Africa (*Niemenak et al., 2008*), it is predicted to have co-opted CS genes for xanthine alkaloid methylation because the order Malvales to which it belongs appears to have lost XMT orthologues prior to its origin (*Figure 8*). Tests of this hypothesis await genomic sequences and functional studies of *Cola* enzymes. However, even with a functional XMT or CS enzyme, gene duplication and protein functional diversification appears to be required to assemble a complete pathway to caffeine as shown here for YM. Nonetheless, because molecular clock analyses indicate that the caffeine-producing *Coffea*, *Camellia*, *Citrus*, *Paullinia*, and *Ilex* lineages each originated within only the last 10–20 million years (*Buerki et al., 2011*; *Hamon et al., 2017*; *Pfeil and Crisp, 2008*; *Yao et al., 2021*; *Zan et al., 2023*), it suggests that the evolution of novel specialized metabolic pathways like that of caffeine can be relatively rapid.

# Materials and methods
## Plant materials
*I. paraguariensis* A. St.-Hil. var. *paraguariensis*, cv CA 8/74 (INTA-EEA Cerro Azul, Misiones, Argentina) and cv SI-49 (Establecimiento Las Marías S.A.C.I.F.A., Corrientes, Argentina) were used in this study. High productivity, increased tolerance to drought, and ease of propagation with stem cuttings characterize these genotypes (*Acevedo et al., 2019*; *Avico et al., 2023*; *Tarragó et al., 2012*).

## DNA extraction and sequencing
Two DNA extraction and sequencing approaches were combined to improve the accuracy of genome assembly. First, young leaves of cv CA 8/74, preserved in silica-gel, were used to isolate total genomic DNA with the DNeasy Plant Mini Kit (QIAGEN), following the manufacturer's instructions. Paired-end libraries (with insert sizes of 350

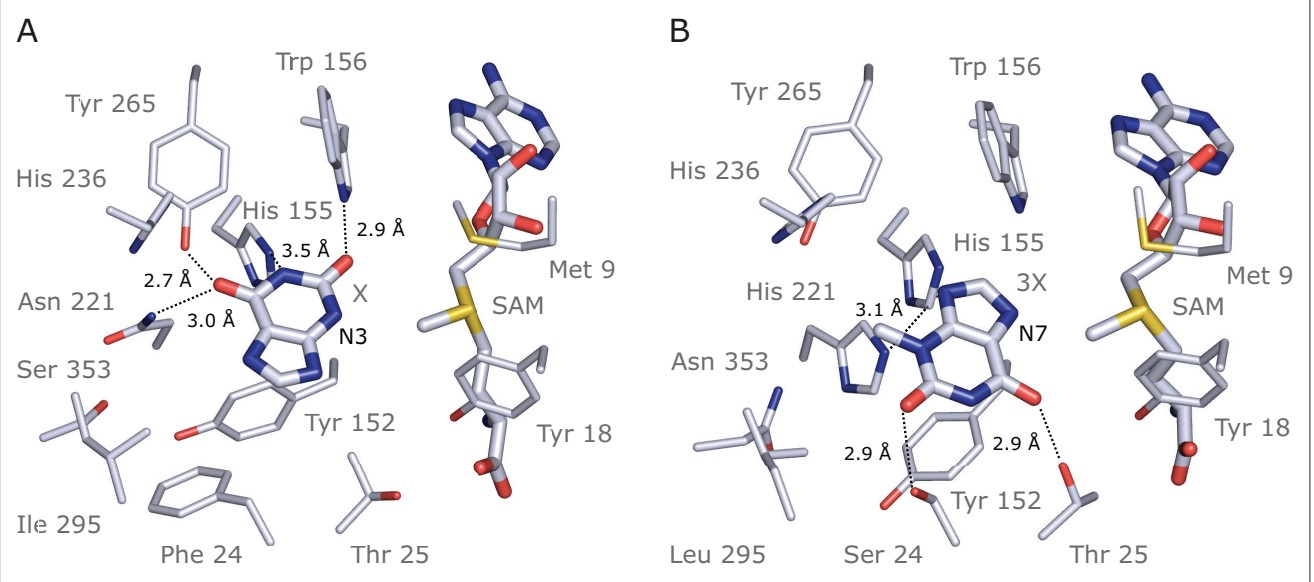

**Figure 7.** Docking models of xanthine alkaloids in IpCS1 and IpCS2 active sites. (**A**) IpCS1–X complex. (**B**) IpCS2–3X complex. Protein residues are displayed as lines with carbon atoms coloured in bluewhite while small molecules – xanthine (X), 3-methylxanthine (3X), caffeine (CF), paraxanthine (PX), S-adenosyl-L-methionine (SAM), and S-adenosyl-homocysteine (SAH) – are drawn as sticks. Colour code for the rest of the atoms: nitrogen (blue), oxygen (red), and sulphur (yellow). Hydrogen bond interactions are indicated as black dotted lines.

The online version of this article includes the following figure supplement(s) for figure 7:

**Figure supplement 1.** AlphaFold2-ColabFold Model Quality assessment of IpCS1, IpCS2, and IpCS3 models.

and 550 bp) and mate-pair libraries (with insert sizes of 3, 8, and 12 kbp) were constructed using the Illumina TruSeq DNA Sample Preparation Kit and Illumina Nextera Mate Pair Library Preparation Kit following the kit's instructions, respectively. The obtained libraries were sequenced on an Illumina HiSeq 1500 platform, generating ~263.2 Gb of raw data. Second, young leaves of cv SI-49, preserved in silica-gel, were used to purify high molecular weight DNA with the Quick-DNA HMW MagBead Kit (Zymo Research), according to the manufacturer's instructions. Long reads libraries were prepared using Sequel Binding Kit 1.0 (Pacific Biosciences), following the manufacturer's instructions. The obtained libraries were subsequently sequenced on PacBio Sequel I (Pacific Biosciences) using Sequel Sequencing Kit 1.0 (Pacific Biosciences) and SMRT Cell 1M (Pacific Biosciences), generating ~77.5 Gb of additional raw data.

## Genome assembly and quality assessment

We opted for a pipeline that could take advantage of both short and long sequencing technologies. For the short reads, we applied Trimmomatic v.0.39 (**Bolger et al., 2014**) to remove adaptor contaminations and filter low-quality reads (reads with mean quality scores ≤25, reads where the quality of the bases at the head or tail was ≤10 and reads with a length ≤30 bp). The resulting clean reads were then corrected using Quake v.0.3 (**Kelley et al., 2010**). Contig assembly and scaffolding was done using the assembler SOAPdenovo v.2 (**Luo et al., 2012**) (55-mer size), with the mate-pair reads being used to link contigs into scaffolds. After the assembly, DeconSeq v.0.4.3 (**Schmieder and Edwards, 2011**) was used to detect and remove sequence contaminants. Contigs and scaffolds clearly belonging to the chloroplast and mitochondria genomes were also discarded. YM transcriptome sequences (**Acevedo et al., 2019**; **Debat et al., 2014**; **Fay et al., 2018**) and public databases KOG (**Tatusov et al., 2003**) and DEG (**Luo et al., 2014**) were used to validate the genome assembly. Canu v.2.2 (**Koren et al., 2017**) was used to perform the self-correction and assembly of the long reads, using the default parameters and stopOnLowCoverage = 20. For both short and long assemblies, we separated the assembly haplotypes (haplotigs) using PurgeHaplotigs (**Roach et al., 2018**) with the recommended parameter values. Then, we merged both SOAPdenovo v.2 and Canu v.2.2 curated assemblies using Quickmerge v.03 (**Chakraborty et al., 2016**), where only contigs with minimum overlap of 5000 bp (-ml 5000) were merged and only the contigs greater than 1000 bp (-l 1000) were retained. The

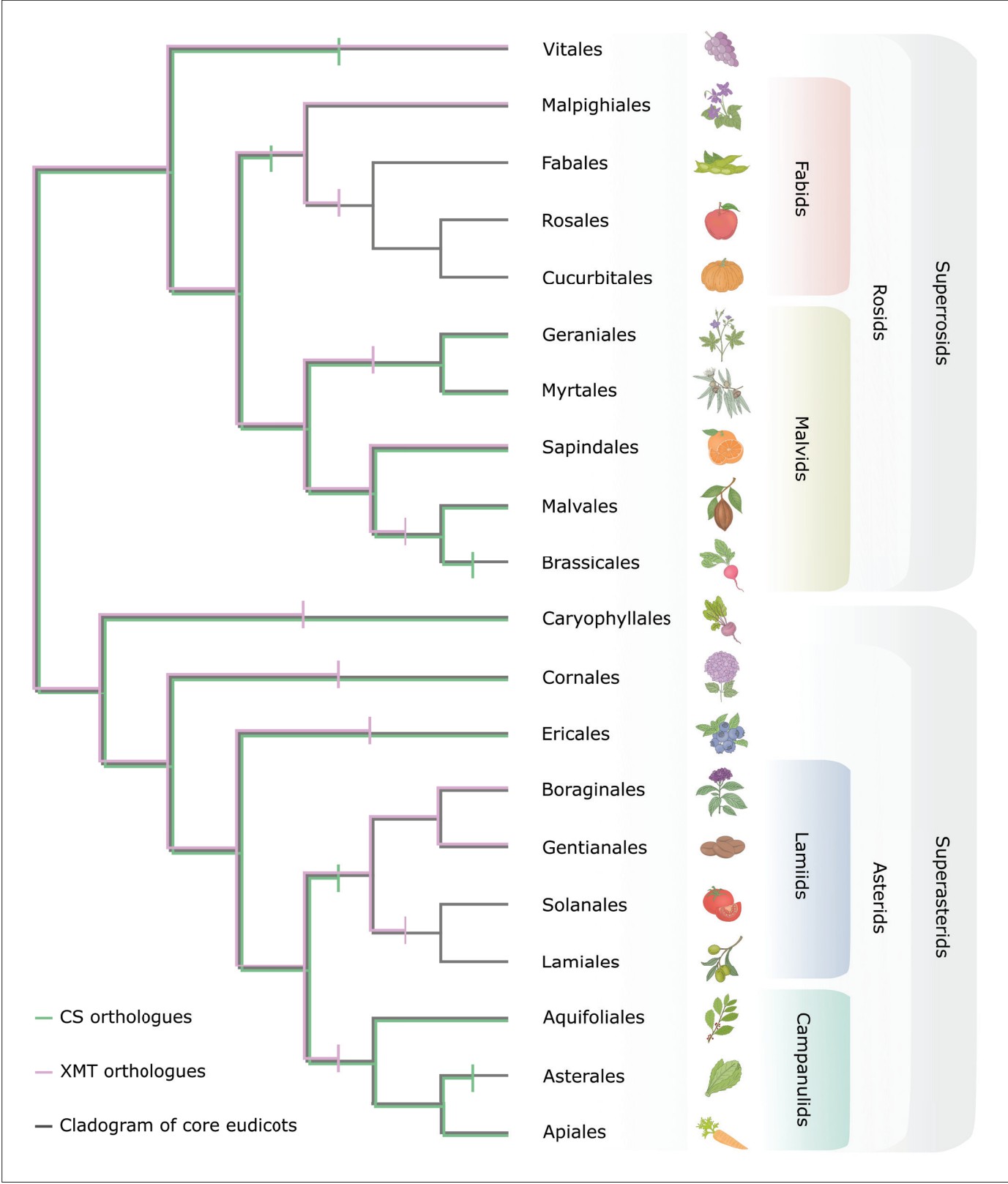

**Figure 8.** Only CS genes were available for co-option and utilization for xanthine alkaloid biosynthesis in yerba mate whereas coffee only had xanthine methyltransferase (XMT) genes. Both CS- and XMT-type caffeine biosynthetic enzymes were present in the ancestor of core eudicots but numerous apparent losses of one or the other or both has occurred during lineage diversification. Gene loss is represented by vertical bar on relevant branches of the cladogram.

*Figure 8 continued on next page*

*Figure 8 continued*

The online version of this article includes the following figure supplement(s) for figure 8:

**Figure supplement 1.** Only CS genes are available for co-option and utilization for xanthine alkaloid biosynthesis in yerba mate.

resulting scaffolds and contigs were refined further with the gap-filling module in SOAPdenovo v.2 (GapCloser) and SSPACE v.2.1.1 (**Boetzer et al., 2011**).

## Gene prediction and annotation

First, we masked the genome assembly with RepeatMasker (http://repeatmasker.org/). Then, we predicted the protein- and non-coding genes using Funannotate v.1.8.13 (**Palmer and Stajich, 2019**) previously training it with the available *I. paraguariensis* RNA-Seq experiments (NCBI projects PRJNA315513, PRJNA375923, and PRJNA251985). Then, Infernal v.1.1.4 (**Nawrocki and Eddy, 2013**) was employed to improve the prediction of small RNAs and microRNAs, while tRNAScan-SE v.2.0 (**Chan and Lowe, 2019**) was used to improve the prediction of transfer RNAs. Ribosomal RNAs were predicted using RNAmmer v.1.2 (**Lagesen et al., 2007**). The TAPIR web server (**Bonnet et al., 2010**) (http://bioinformatics.psb.ugent.be/webtools/tapir) and the TargetFinder software v.1.7 (**Fahlgren and Carrington, 2010**) were used to identify miRNA targets. InterProScan v.5.55-88.0 (**Jones et al., 2014**) and eggNOG-mapper v.2.1.7 (**Huerta-Cepas et al., 2019**) were employed for the functional assignment of the predicted genes.

## Repeat content estimation

The repeat content was estimated employing Dfam TE Tools v.1.5 (https://github.com/Dfam-consortium/TETools copy archived at **Rosen and Gray, 2024**). First, we used RepeatModeler v.2.0.3 (**Flynn et al., 2020**) to build a database with *Ilex* repeat families. Then, we merged that database with Dfam v3.6 (**Hubley et al., 2016**) and GIRI Repbase ver 20181026 (**Jurka et al., 2005**). Finally, we ran Repeat-Masker on the assembly using the merged database to look for repeat sequences.

## Genome duplication analysis

Rates of synonymous substitution ($K_s$) between paralogous genes and orthologous genes in *Lactuca sativa*, *Daucus carota*, *I. paraguariensis*, *C. canephora*, and *Vitis vinifera* were determined using CoGe's tool SynMap (https://genomevolution.org/). Gaussian mixture models were fitted to the resulting $K_s$ distributions with the mclust R package v.5.0 (**Scrucca et al., 2016**), and significant peaks were identified using the SiZer R package v.0.1-7 (**Chaudhuri and Marron, 2000**). To estimate the age of the lineage-specific polyploidization event (Ip-α) in *Ilex*, we considered two different phylogenies (a multiple nuclear genome phylogeny and a plastid genome phylogeny). With the median $K_s$ value of YM-grape orthologues (~0.89) and the divergence date of the two species (125.64 Ma for the multiple nuclear genome phylogeny and 123.7 Ma for the plastid genome phylogeny), we calculated the number of substitution per synonymous site per year ($r$) for YM (divergence date = $K_s$/($2 \times r$)). Conforming to the multiple nuclear genome phylogeny, the YM $r$ value is 3.54E−9; while for the plastid genome phylogeny, the YM $r$ value is 3.59E−9. These $r$ values and the SiZer $K_s$ range of YM paralogues (~0.35–0.5) were then applied to estimate the age of Ip-α. Finally, to determine the syntenic depth ratio between *I. paraguariensis* and *C. canephora* and *V. vinifera*, we employed CoGe's tool SynFind (https://genomevolution.org/), using a distance cutoff of 10 genes and requiring at least 5 gene pairs per synteny block.

## Gene expression quantitation

First, YM transcriptome reads (PRJNA315513) were mapped to IpCS1–5 transcripts, obtained from the de novo transcriptome assembly and annotation, using BWA (**Li and Durbin, 2009**). Then, with the number of mapped reads, the abundance of each transcript was calculated, normalized by transcript length and transcriptome size (quantification in RPKM, reads per kilobase per million mapped reads).

## Cloning, mutagenesis, heterologous expression, and purification of enzymes

Two different approaches were used to clone IpCS genes: RT-PCR from leaf tissue and custom gene synthesis. For RT-PCR of IpCS2, cDNA was obtained from 1 µg of RNA from fresh YM leaves using standard procedures and cycling conditions with the following two primers: IpCS2F 5′-ATGGACGTGAAG GAAGCAC-3′ and IpCS2R 5′-CTATCCCATGGTCCTGCTAAG-3′. Following amplification, cDNA was cloned using the pTrcHis TOPO TA Expression Kit (Invitrogen, Carlsbad, CA). Ligation of cDNA into the pTrcHis vector and subsequent transformation into Top10 *E. coli* cells was carried out according to the manufacturer's protocol. The transformation mixture was incubated overnight at 37°C on LB plates containing 50 µg/ml ampicillin. Colonies were screened by PCR to obtain full-length inserts that were subsequently verified for insert orientation by DNA sequencing. For IpCS1 and 3 and AncIpCS1 and 2, gene sequences were synthesized by GenScript with codons optimized for expression in *E. coli*. Synthesized genes were subcloned from the pUC57 cloning vector into the pET-15b (Novagen) expression vector using 1.5 µg of DNA and NdeI and BamHI in 30 µl reactions. Linear fragments corresponding to the expected sizes were gel purified using the QIAEX II Gel Extraction Kit (QIAGEN Corp) according to the manufacturer's instructions. Purified DNA fragments were ligated into pET-15b using T4 DNA ligase from New England Biolabs. Ligation products were transformed into Top10 *E. coli* cells using 2 µl of the ligation reaction. Site-directed mutagenesis of AncIpCS2 was carried out using the Agilent QuikChange Lightning Kit (Agilent Technologies Inc, Santa Clara, CA) following the manufacturer's protocol. Minipreps of positive transformants were obtained using a QIAprep Spin Miniprep Kit (QIAGEN Corp) and 10 ng of each plasmid was used to transform BL21 *E. coli* cells using standard plating and incubation methods.

Induction of $His_6$-protein was achieved in 50 ml cell cultures of BL21 (DE3) with IpCS1 and 3 and AncIpCS1 and 2 in pET-15b or Top10 with IpCS2 in pTrcHis with the addition of 1 mM IPTG at 23°C for 6 hr. Purification of the $His_6$-tagged protein was achieved by TALON spin columns (Takara Bio) following the manufacturer's instructions. Bradford assays were used to determine purified protein concentration, and recombinant protein purity was evaluated on sodium dodecyl sulphate–polyacrylamide gel electrophoresis gels.

## Enzyme assays

All enzymes were tested for activity with the eight xanthine alkaloid substrates shown in *Figure 1*. Radiochemical assays were performed at 24°C for 60 min in 50 µl reactions that included 50 mM Tris–HCl buffer, 0.01 µCi (0.5 µl) $^{14}$C-labelled SAM, 10–20 µl purified protein, and 1 mM methyl acceptor substrate dissolved in 0.5 M NaOH. Negative controls were composed of the same reagents, except that the methyl acceptor substrate was omitted and 1 µl of 0.5 M NaOH was added instead. Methylated products were extracted in 200 µl ethyl acetate and quantified using a liquid scintillation counter. The highest enzyme activity reached with a specific substrate was set to 1.0 and relative activities with remaining substrates were calculated. Each assay was run at least three times so that mean, plus standard deviation, could be calculated.

## High-performance liquid chromatography

Product identity of enzyme assays was determined using high-performance liquid chromatography (HPLC) on 500 µl scaled-up reactions utilizing all the same reagents as described above except that non-radioactive SAM was used as the methyl donor and reactions were allowed to progress for 4 hr. Whole reactions were filtered through Vivaspin columns (Sartorius) to remove all protein prior to injection in the HPLC. Mixtures were separated by HPLC using a two-solvent system with a 250 mm × 4.6 mm Kinetex 5 µM EVO C18 column (Phenomenex). Solvent A was 99.9% water with 0.1% trifluoroacetic acid and Solvent B was 80% acetonitrile, 19.9% water and 0.1% TFA and a 0–20% gradient was generated over 16 min with a flow rate of 1.0 ml/min. Product identity was determined by comparing retention times and absorbance at 254 and 272 nm of authentic standards. Reactions were compared to negative controls in which no methyl acceptor substrates were added.

## Phylogenetic analyses

In order to accurately determine the orthology of YM SABATH sequences encoded in the genome, we compared them to all previously functionally characterized gene family members in other species.

We also included CS and XMT orthologues from the orders of caffeine-producing species (Malvales, Ericales, Gentianales, Sapindales) available in public databases (GenBank, OneKP) as shown in *Figure 3*. Accession numbers for all sequences are provided in . Alignment of amino acid sequences was achieved using MAFFT v.7.0 (*Katoh and Standley, 2013*) and employing the auto search strategy to maximize accuracy and speed. A phylogenetic estimate was obtained using FastTree v.2 (*Price et al., 2010*) assuming the Jones–Taylor–Thorton model of amino acid substitution with a CAT approximation using 20 rate categories. Reliability of individual nodes was estimated from local support values using the Shimodaira–Hasegawa test as implemented in FastTree.

## Ancestral sequence resurrection

In order to obtain accurate ancestral CS protein estimates, we assembled two datasets to assess variation in terms of sampling. The first dataset included 154 sequences including all CS-type enzymes we could retrieve from GenBank and China National Gene Bank as well as representatives of all other functionally characterized clades of SABATH enzymes (*Figure 5—figure supplement 1*). In this dataset, the only *Ilex* sequences available were IpCS1–3. This dataset resulted in highly confident estimates for AncIpCS1 and AncIpCS2 (average site-specific posterior probabilities >0.99 in both cases). Subsequently, once additional *Ilex* genomes became available, we estimated a second set of ancestral sequences using 29 CS-type enzymes from asterids to assess uncertainty in our initial estimates (*Figure 5—figure supplement 2*). In this subsequent analysis, highly confident estimates for AncIpCS1 and AncIpCS2 were obtained with average site-specific posterior probabilities >0.99 in both cases (see *Figure 5*). MAFFT v.7.0 (*Katoh and Standley, 2013*) was used to align the protein sequences in both datasets using the auto search strategy to maximize accuracy and speed; subsequently, IQTree (*Trifinopoulos et al., 2016*) was used to obtain trees describing the relationships amongst the aligned sequences for both datasets. For the first set of ancestral sequence estimates, the Jones, Taylor, and Thorton matrix model for amino acid substitution and the Free rate model for among-site rate heterogeneity (*Yang, 1995*) was determined to be the best fit. For the second dataset, the Q matrix as estimated for plants (*Ran et al., 2018*) with a gamma model for rate heterogeneity was the preferred model. IQTree estimates ancestral sequences using the empirical Bayesian approach (*Trifinopoulos et al., 2016*). In order to determine ancestral protein lengths in regions with alignment gaps, we coded each gap for the number of amino acids possessed and used parsimony to determine ancestral residue numbers as in our previous studies (*Huang et al., 2016*). The estimated sequences were synthesized by Genscript Corp and had codons chosen for optimal protein expression in *Escherichia coli* and were cloned into pET15b for expression and purification using the His$_6$ tag. Details of expression were the same as described above for the modern-day enzymes. Although the two separate ancestral sequence estimates are highly similar to one another (>95% identity in both cases), the two AncIpCS1 proteins differ at 10 positions and those for AncIpCS2 differ at seven positions (*Figure 5—figure supplements 3 and 4*).

## Crystallization, data collection, phasing, and refinement of IpCS3

Initial crystallization screening was performed using the IpCS3 methyltransferase at a concentration of 30 mg/ml incubated with 2 mM TB and 2 mM SAM. Sitting-drop for crystallization screening was set up by equal volume of precipitant and protein solution (0.25:0.25 µl) using a Crystal Gryphon robot (Art Robbins Instruments) and a reservoir volume of 45 µl. Trays were incubated at 9°C. Initial hits were further optimized using the hanging-drop method at 9°C, with 150 µl reservoir solution and 1:1 ratio of precipitant to protein and ligand solution in a 2-µl drop. Attempts to crystallize with SAH or uncleavable SAM analogs and TB to attain a pre methylation structure were unsuccessful given the poor diffraction of these crystals. Therefore, the latter was composed of 33 mg/ml IpCS3 protein concentration, 4 mM TB and 2 mM SAM, and the crystallization condition was optimized to 25% PEG 3350, 0.2 M NH$_4$SO$_4$, 0.1 M Bis-Tris methane pH 5.5. Square crystals grew over 10 days, but initial X-ray crystallography data revealed a poor electron density for SAH and an electron density in the active site for CF, the product, rather than for TB. Consequently, crystals were grown in the aforementioned condition and subsequently soaked for 4 hr at 9°C in the precipitant solution supplemented with 10 mM SAH and 10 mM TB. The idea was to supply an excess of the expended methyl source and additional TB to convert any existing SAM as we did not have access to caffeine as a reagent. Crystals were transiently soaked in the precipitant solution supplemented with 20% ethylene

glycol immediately prior to vitrification by direct immersion into liquid nitrogen. Diffraction data were collected at the Advanced Photon Source (APS) at Argonne National Laboratory Sector-21 via the Life Sciences-Collaborative Access Team (LS-CAT) at beamline 21-ID-G. Diffraction data were indexed, integrated, and scaled using the autoPROC software package (*Vonrhein et al., 2011*). The structure was solved by molecular replacement using Phaser-MR included in the Phenix software package (*Adams et al., 2010*), using PDB ID 6LYH structure as the replacement model. The model was subject to rounds of manual building followed by refinement using REFMAC5 (*Murshudov et al., 2011*), and was manually built in COOT v.0.9.8.3 (*Emsley et al., 2010*). Crystallographic statistics are listed in *Table 5*.

## Structure prediction and molecular docking

Protein structures of IpCS enzymes were predicted using the ColabFold implementation of Alpha-Fold2 (*Mirdita et al., 2022*) with no template. Diagnostic plots depicting the MSA coverage, alignment error and LDDT are shown in the supplementary information (*Figure 7—figure supplement 1*). Structures of xanthine alkaloid ligands (X, 3X, and TB) were downloaded from the ChEMBL database *Mendez et al., 2019*; protonation states were checked by Chemicalize (*Swain, 2012*) and optimized using the VMD Molefacture plugin (*Humphrey et al., 1996*). The receptor structures were prepared following the standard AutoDock protocol (*Morris et al., 2009*) using the prepare_receptor4.py script from AutoDock Tools. All non-polar hydrogens were merged, and Gasteiger charges and atom types were added. The ligand PDBQT was prepared using the prepare_ligand4.py script. The grid size and position were chosen to contain the whole ligand-binding site (including all protein atoms closer than 5 Å from all ligands). For each system, 10 different docking runs were performed. Docking was performed using AutoDock Vina v.1.2.0 (*Eberhardt et al., 2021*). The docking results were further analysed by visual inspection. Images of the molecules were prepared using the PyMOL molecular graphics system (*Schrodinger, 2015*).

## Synteny comparisons and phylogenetic distribution of CS and XMT

The presence or absence of CS and XMT genes was determined for orders of plants for which at least one genomic sequence exists, as shown in *Figure 8*. For those species which do not yet have an available assembly, we used BLAST (*Altschul et al., 1990*) analyses of GenBank (nr and TSA databases), Phytozome (*Goodstein et al., 2012*) as well as the OneKP dataset (*One Thousand Plant Transcriptomes Initiative, 2019*) in China National GeneBank. BLAST combined with subsequent phylogenetic analyses were also used to verify presence/absence of CS- or XMT-type sequences in cases where the syntenic regions did not appear to encode one or the other gene. Comparisons of the CS and XMT syntenic regions were performed using CoGe's tool GEvo (https://genomevolution.org/).

# Acknowledgements

This work was supported by Consejo Nacional de Investigaciones Científicas y Técnicas de Argentina (CONICET); PRO.MATE.AR project, funded by Secretaría de Políticas Universitarias del Ministerio de Educación de la Nación Argentina; CABANA project, funded by UKRI-BBSRC on behalf of the Global Challenges Research Fund (BB/P027849/1), the U.S. National Science Foundation (NSF) (grants MCB-1120624 and MCB-2325341 to T.J.B.), the Lee Honors College at WMU (to M.S. and T.J.B.), and the European Molecular Biology Laboratory (EMBL). We would like to express our gratitude to Centro de Cómputos de Alto Rendimiento (CeCAR) and UBA-FCEN-QB-Cluster for providing access to computational resources, which facilitated the majority of computational analyses in this work. Special thanks are extended to Kevin Blair and the WMU Department of Chemistry for facilitating our HPLC analyses.

## Additional information

### Funding

| Funder | Grant reference number | Author |
|---|---|---|
| Consejo Nacional de Investigaciones Científicas y Técnicas | | Adrian G Turjanski |
| Biotechnology and Biological Sciences Research Council | BB/P027849/1 | Adrian G Turjanski |
| National Science Foundation | MCB-1120624 | Todd J Barkman |
| European Molecular Biology Laboratory | | Federico A Vignale |
| Ministerio de Educación de la Nación | PRO.MATE.AR | Adrian G Turjanski |
| Lee Honors College, Western Michigan University | | Madeline N Smith |
| National Science Foundation | MCB-2325341 | Todd J Barkman |

The funders had no role in study design, data collection and interpretation, or the decision to submit the work for publication.

### Author contributions

Federico A Vignale, Conceptualization, Data curation, Software, Formal analysis, Supervision, Validation, Investigation, Visualization, Methodology, Writing – original draft, Project administration, Writing – review and editing; Andrea Hernandez Garcia, Data curation, Software, Formal analysis, Validation, Investigation, Visualization, Writing – review and editing; Carlos P Modenutti, Conceptualization, Resources, Data curation, Software, Formal analysis, Investigation, Methodology, Project administration, Writing – review and editing; Ezequiel J Sosa, Renato Oliveira, Gisele L Nunes, Data curation, Software, Formal analysis, Validation, Investigation, Methodology, Writing – review and editing; Lucas A Defelipe, Software, Formal analysis, Investigation, Visualization, Methodology, Writing – review and editing; Raúl M Acevedo, Resources, Investigation, Methodology, Writing – review and editing; German F Burguener, Data curation, Software, Formal analysis, Methodology; Sebastian M Rossi, Formal analysis, Investigation, Writing – review and editing; Pedro D Zapata, Dardo A Marti, Pedro Sansberro, Guilherme Oliveira, Satish Nair, Funding acquisition; Emily M Catania, Madeline N Smith, Nicole M Dubs, Formal analysis, Investigation; Todd J Barkman, Conceptualization, Resources, Data curation, Software, Formal analysis, Supervision, Funding acquisition, Validation, Investigation, Visualization, Methodology, Writing – original draft, Project administration, Writing – review and editing; Adrian G Turjanski, Conceptualization, Resources, Supervision, Funding acquisition, Project administration, Writing – review and editing

### Author ORCIDs

Federico A Vignale ![ORCID] http://orcid.org/0000-0003-0849-0916
Lucas A Defelipe ![ORCID] https://orcid.org/0000-0001-7859-7300
Raúl M Acevedo ![ORCID] https://orcid.org/0000-0001-7582-3018
German F Burguener ![ORCID] https://orcid.org/0000-0002-8600-7136
Sebastian M Rossi ![ORCID] https://orcid.org/0000-0002-6694-0076
Pedro D Zapata ![ORCID] https://orcid.org/0000-0001-6476-8324
Pedro Sansberro ![ORCID] https://orcid.org/0000-0002-6540-3666
Nicole M Dubs ![ORCID] https://orcid.org/0009-0006-0695-398X
Todd J Barkman ![ORCID] http://orcid.org/0000-0003-2259-2345
Adrian G Turjanski ![ORCID] https://orcid.org/0000-0003-2190-137X

Decision letter and Author response
Decision letter https://doi.org/10.7554/eLife.104759.sa1
Author response https://doi.org/10.7554/eLife.104759.sa2

## Additional files

### Supplementary files
MDAR checklist

### Data availability
The Illumina and PacBio raw sequence data, assembly and annotation were deposited in the European Nucleotide Archive (ENA) under BioProject No. PRJEB65927. An assembly obtained only with the Illumina data was also deposited in ENA under BioProject No. PRJEB36685. The plasmids used to produce proteins are freely available upon request. The atomic coordinates and structure factors have been deposited in the Protein Data Bank, Research Collaboratory for Structural Bioinformatics, Rutgers University, New Brunswick, NJ (http://www.rscb.org) with the accession code 8UZD for the IpCS3 structure bound to caffeine and SAH.

The following datasets were generated:

| Author(s) | Year | Dataset title | Dataset URL | Database and Identifier |
| --- | --- | --- | --- | --- |
| Vignale FA | 2024 | Ilex paraguariensis var. paraguariensis genome (Illumina and PacBio) | https://www.ebi.ac.uk/ena/browser/view/PRJEB65927 | ENA, PRJEB65927 |
| Vignale FA | 2024 | Ilex paraguariensis var. paraguariensis genome (Illumina) | https://www.ebi.ac.uk/ena/browser/view/PRJEB36685 | ENA, PRJEB36685 |
| Hernandez Garcia A | 2024 | The structure of IpCS3, a theobromine methyltransferase from Yerba Mate | https://www.rcsb.org/structure/8UZD | RCSB Protein Data Bank, 8UZD |

The following previously published datasets were used:

| Author(s) | Year | Dataset title | Dataset URL | Database and Identifier |
| --- | --- | --- | --- | --- |
| Fay JV | 2016 | Ilex paraguariensis multiple library de novo transcriptome assembly | https://www.ebi.ac.uk/ena/browser/view/PRJNA315513 | ENA, PRJNA315513 |
| Debat HJ | 2014 | Yerba mate (Ilex paraguariensis St. Hil.) NGS DN Transcriptome assembly | https://www.ncbi.nlm.nih.gov/sra/SRP043293 | NCBI Sequence Read Archive, SRP043293 |
| Acevedo RM | 2019 | RNA-Seq of Ilex paraguariensis: roots and mature leaves | https://www.ncbi.nlm.nih.gov/sra/?term=SRP110129 | NCBI Sequence Read Archive, SRP110129 |

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

## Appendix 1

### Non-coding RNAs in yerba mate

#### 1.1 Transfer RNAs

Analysis of transfer RNA (tRNA) family members revealed the presence of 815 tRNA genes including 726 standard tRNAs, 76 pseudo tRNAs, 11 tRNAs with undetermined isotypes, and 2 possible suppressor tRNAs (*Appendix 1—table 1*). No selenocysteine tRNA was found in the yerba mate genome, which is consistent with the results of other tRNA analyses carried out in higher plants (*Santesmasses et al., 2017*). According to evolutionary analyses, selenoproteins and selenocysteine insertion sequence (SECIS) elements present in protozoans and animals evolved early, and were independently lost in higher plants and fungi through evolution (*Novoselov et al., 2002*). With regard to the nonsense suppressor tRNAs, only ochre and opal nonsense suppressor tRNA genes were found in the yerba mate genome, which suppress the phenotypes of ochre and opal mutations, respectively.

The length of the standard tRNA sequences ranged from 61 to 236 nucleotides, encoding 53 different anti-codons/isoacceptors in total, with tRNA$^{Ser}$ having the highest abundance of genes and tRNA$^{Tyr}$ having the lowest (*Appendix 1—table 1*). In our study, we also found that yerba mate tRNAs contain introns in 13 of the 55 tRNA$^{Met}$, 9 of the 17 tRNA$^{Tyr}$, 1 of the 65 tRNA$^{Ser}$, and 1 of the 22 tRNA$^{Ile}$ genes, providing additional demonstration that tRNA$^{Met}$ and tRNA$^{Tyr}$ are not the only intron containing tRNAs in the plant kingdom as it was previously believed (*Michaud et al., 2011*). The length of these introns varied from 6 to 163 nucleotides and all of them were found at the canonical position, one nucleotide 3′ to the anti-codon loop.

#### 1.2 Ribosomal RNAs

Analysis of ribosomal RNA (rRNA) family members showed the presence of 425 5S rRNA, 23 18S rRNA, and 23 25S rRNA genes. The large variability in the copy number of the rRNA genes has been observed and studied in plants for decades (*Rogers and Bendich, 1987*). It is believed that a high copy number of these genes is important to ensure increased demand of proteosynthesis during plant development, but also to stabilize the cell nucleus (*Garcia et al., 2012*). With regard to its genomic organization, the 18S and 25S rRNA genes were clustered together, while the 5S rRNA genes were tandemly located elsewhere in the genome (S-type arrangement). However, some 5S rRNA genes were linked with the 18S and 25S rRNA genes as well (L-type arrangement). Given this observation, we should incorporate the yerba mate genome to the list of eukaryotic genomes with an L-arrangement of ribosomal DNA.

#### 1.3 Small RNAs

Analysis of small RNA family members revealed the presence of 348 small nuclear RNA (snRNA) genes including 65 U1 snRNA, 46 U2 snRNA, 29 U4 snRNA, 33 U5 snRNA, and 146 U6 snRNA genes corresponding to the major spliceosome complex, and 19 U6atac snRNA, 7 U11 snRNA, and 1 U12 snRNA genes corresponding to the minor spliceosome complex. Furthermore, it showed the presence of 2670 small nucleolar RNA (snoRNA) genes, of which 2631 (~98.54%) were box C/D snoRNA genes and 39 (~1.46%) were box H/ACA snoRNA genes. Both groups of snoRNAs are involved in the cleavage of precursor ribosomal RNA (pre-rRNA) and determine site-specific modification in pre-rRNAs and snRNAs, though the box C/D snoRNAs are usually associated with 2′-*O*-ribose methylation, while the box H/ACA snoRNAs are normally associated with 2′-*O*-ribose pseudouridylation (*Brown et al., 2003*; *Hari and Parthasarathy, 2019*). The greater abundance of the box C/D snoRNAs in the yerba mate genome could be explained, first, by the fact that plants have higher numbers of 2′-*O*-ribose methylated nucleotides than archaea, yeast, and other higher eukaryotes, and second, by the fact that computer algorithms still find difficult to predict the relatively short conserved sequences of box H/ACA snoRNAs (*Brown et al., 2003*). It was remarkable the high copy number (2377) of snoRNA R71 in the yerba mate genome, which is a member of the Box C/D family. This snoRNA, which is thought to function as a 2′-*O*-ribose methylation guide for 18S rRNA, has been identified in multiple copies in most eudicot genomes (*Kiss-László et al., 1996*).

## 1.4 Micro RNAs

Analysis of micro RNA (miRNA) family members revealed the presence of 226 miRNA genes belonging to 30 families. The most abundant miRNAs usually involved in growth and development were miR156, miR166, and miR159, while the most abundant miRNAs normally involved in stress responses were miR169_2, miR167_1, miR399, and miR395 (*Appendix 1—table 2*). Nevertheless, the functions of miRNAs slightly differ among plants. Therefore, to better understand the regulatory effect of miRNAs in yerba mate, we used the TAPIR web server (*Bonnet et al., 2010*) and the TargetFinder softwareto identify yerba mate miRNA targets (*Appendix 1—table 3*). The results obtained allowed us to infer the role of 14 of the 30 miRNA families found in the yerba mate genome. Apparently, miR159, miR164, miR169_2, and miR169_5 regulate a variety of processes related to development and stress responses. On the one hand, both miR159 and miR164 regulate the expression of myb-like transcription factors, involved in auxin homeostasis, lateral root and leaf development, leaf senescence, response to abscisic acid, response to the absence of light and response to salt stress. miR164 also regulates the synthesis of vitamin B5 involved in embryo development. On the other hand, both miR169_2 and miR169_5 regulate the expression of a galactinol synthase involved in the response to cold stress, heat stress, oxidative stress, salt stress and water deprivation; a kinesin-like protein involved in pollen development and a mitogen-activated protein kinase involved in directing cellular responses to mitogens, osmotic stress, heat shock, and proinflammatory cytokines. miR167_1 is probably involved only in plant growth and development as it regulates the expression of an auxin response factor (arf). And last, miR171_1 and miR390 are likely involved only in stress responses. miR171_1 regulates the expression of an RNA-binding family protein and an endoglucanase involved in host defence, whereas miR390 regulates the expression of a rotamase FKBP 1 involved in the response to heat stress, osmotic stress, and wounding. It is important to mention that the functions of all the predicted targets were gathered from the *Arabidopsis* Information Resource (TAIR) (*Rhee et al., 2003*), and therefore the functional involvement of these miRNAs in yerba mate must be experimentally validated.

**Appendix 1—table 1.** Detail of yerba mate tRNA and anti-codon nucleotide sequences.

| tRNA genes | Anti-codon counts | | | | | | Total No. of tRNAs |
|---|---|---|---|---|---|---|---|
| POLAR | | | | | | | |
| Asparagine (Asn) | GTT (36) | ATT (0) | | | | | 36 |
| Cysteine (Cys) | GCA (22) | ACA (0) | | | | | 22 |
| Glutamine (Gln) | TTG (13) | CTG (10) | | | | | 23 |
| Glycine (Gly) | GCC (32) | TCC (11) | CCC (8) | ACC (0) | | | 51 |
| Serine (Ser) | GCT (20) | TGA (20) | AGA (15) | CGA (5) | GGA (5) | ACT (0) | 65 |
| Threonine (Thr) | TGT (11) | AGT (16) | GGT (6) | CGT (2) | | | 35 |
| Tyrosine (Tyr) | GTA (17) | ATA (0) | | | | | 17 |
| NON-POLAR | | | | | | | |
| Alanine (Ala) | AGC (12) | CGC (4) | TGC (11) | GGC (0) | | | 27 |
| Isoleucine (Ile) | AAT (14) | TAT (6) | GAT (2) | | | | 22 |
| Leucine (Leu) | CAA (23) | AAG (10) | CAG (4) | TAG (8) | TAA (6) | GAG (0) | 51 |
| Methionine (Met) | CAT (55) | | | | | | 55 |
| Phenylalanine (Phe) | GAA (30) | AAA (2) | | | | | 32 |
| Proline (Pro) | AGG (10) | TGG (28) | CGG (4) | GGG (0) | | | 42 |
| Tryptophan (Trp) | CCA (31) | | | | | | 31 |
| Valine (Val) | AAC (11) | GAC (10) | CAC (9) | TAC (7) | | | 37 |
| POSITIVELY CHARGED | | | | | | | |
| Arginine (Arg) | ACG (15) | TCT (14) | CCT (7) | CCG (6) | TCG (6) | GCG (3) | 51 |
| Histidine (His) | GTG (25) | ATG (2) | | | | | 27 |
| Lysine (Lys) | CTT (10) | TTT (17) | | | | | 27 |

*Appendix 1—table 1 Continued on next page*

*Appendix 1—table 1 Continued*

| tRNA genes | Anti-codon counts | | | Total No. of tRNAs |
|---|---|---|---|---|
| **NEGATIVELY CHARGED** | | | | |
| Aspartic acid (Asp) | GTC (39) | ATC (1) | | 40 |
| Glutamic acid (Glu) | CTC (14) | TTC (21) | | 35 |
| Selenocysteine tRNAs | TCA (0) | | | 0 |
| Possible suppressor tRNAs | CTA (0) | TTA (1) | TCA (1) | 2 |
| tRNAs with undetermined isotypes | | | | 11 |
| Predicted pseudogenes | | | | 76 |

**Appendix 1—table 2.** miRNA families predicted in the yerba mate genome.

| miRNA | Functional involvement in other eudicot plants |
|---|---|
| | Seed growth and development *Chi et al., 2011*; *Song et al., 2011* |
| | Fruit development (*Pantaleo et al., 2010*) |
| miR156 | Drought/cold stress (*Curaba et al., 2012*; *Zhu and Luo, 2013*) |
| | Growth and development (*Varkonyi-Gasic et al., 2010*) |
| | Phase change from vegetative to reproductive growth (*Han et al., 2014*) |
| | Lipid and protein accumulation (*Zhao et al., 2010*) |
| miR159 | Drought stress (*Barrera-Figueroa et al., 2011*) |
| | Growth and development (*Gu et al., 2013*; *Wang et al., 2011*) |
| | Fibrous root and storage root development (*Sun et al., 2015*) |
| miR160 | Drought stress (*Nadarajah and Kumar, 2019*) |
| miR162_2 | Storage root initiation and development (*Sun et al., 2015*) |
| | Lateral root and leaf development (*Deng et al., 2015*) |
| | Fibrous root and storage root development (*Sun et al., 2015*) |
| | Seed development (*Song et al., 2011*) |
| miR164 | Drought stress (*Ferreira et al., 2012*) |
| | Seed development (*Song et al., 2011*) |
| | Fibrous root and storage root development (*Sun et al., 2015*) |
| | Drought stress (*Barrera-Figueroa et al., 2011*) |
| miR166 | Disease resistance (*Guo et al., 2011*) |
| | Growth and development (*Varkonyi-Gasic et al., 2010*) |
| miR167_1 | Drought/cold stress (*Barrera-Figueroa et al., 2011*; *Jeong et al., 2011*) |
| | Development (*Gu et al., 2013*) |
| miR168 | Resistance to fire blight (*Kaja et al., 2015*) |
| miR169_2; miR169_5 | Drought/cold/salt stress (*Carnavale Bottino et al., 2013*; *Koc et al., 2015*; *Sheng et al., 2015*; *Shui et al., 2013*) |
| miR171_1; miR171_2 | Development (*Chaves et al., 2015*; *Zhang et al., 2011*) |
| | Lipid and protein accumulation (*Zhao et al., 2010*) |

*Appendix 1—table 2 Continued on next page*

*Appendix 1—table 2 Continued*

| miRNA | Functional involvement in other eudicot plants |
|---|---|
| | Development (*Sun et al., 2012*) |
| | Starch biosynthesis (*Chen et al., 2015*) |
| miR172 | Drought/cold stress (*Koc et al., 2015*) |
| | Drought stress (*Shui et al., 2013*) |
| miR390 | Leaf morphology (*Karlova et al., 2013*) |
| miR394 | Drought/salt stress (*Song et al., 2013*) |
| miR395 | Low sulfate response (*Katiyar et al., 2012*) |
| | Seed development (*Gao et al., 2015*) |
| | Starch biosynthesis (*Chen et al., 2015*) |
| miR396 | Drought/salt stress (*Shui et al., 2013*; *Xie et al., 2014*) |
| miR397 | Drought/cold stress (*Koc et al., 2015*) |
| | Fibrous root and storage root development (*Sun et al., 2015*) |
| miR398 | Salt stress (*Carnavale Bottino et al., 2013*) |
| | Phosphate homeostasis (*Katiyar et al., 2012*; *Pant et al., 2008*) |
| miR399 | Shoot to root transport (*Pant et al., 2008*) |
| miR403 | Drought stress (*Shui et al., 2013*) |
| miR405 | Transposon derived (*Xie et al., 2005*) |
| | Tolerance to Boron deficiency (*Lu et al., 2015*) |
| | Cold stress (*Zhang et al., 2014*) |
| miR408 | Response to wounding and topping (*Tang et al., 2012*) |
| | Metabolism (*Din et al., 2014*) |
| miR473 | Stress response (*Patanun et al., 2013*) |
| miR474 | Drought stress (*Kantar et al., 2011*) |
| miR475 | Metabolism (*Din et al., 2014*) |
| miR477 | Starch biosynthesis (*Xie et al., 2011*) |
| miR530 | Disease resistance (*Zhao et al., 2015*) |
| miR1023 | Disease resistance (*Jiao and Peng, 2018*) |
| miR1446 | Stress response (*Lu et al., 2008*) |

**Appendix 1—table 3.** miRNA targets predicted in the yerba mate genome.

| Targets IDs | Description | miR159 | miR164 | miR167_1 | miR168 | miR169_2 | miR169_5 | miR171_1 | miR171_2 | miR390 | miR394 | miR396 | miR397 | miR398 | miR403 |
|---|---|---|---|---|---|---|---|---|---|---|---|---|---|---|---|
| ILEXPARA_008283 | Uncharacterized protein | * | | | | | | | | | | | | | |
| ILEXPARA_029002 | Uncharacterized protein | * | | | | | | | | | | | | | |
| ILEXPARA_031381 | Uncharacterized protein | * | | | | | | | | | | | | | |
| ILEXPARA_043376 | Uncharacterized protein | * | | | | | | | | | | | | | |
| ILEXPARA_013180 | *ileS*, isoleucine tRNA ligase | * | | | | | | | | | | | | | |
| ILEXPARA_000910 | myb-like transcription factor | * | * | | | | | | | | | | | | |
| ILEXPARA_028644 | Uncharacterized protein | | * | | | | | | | | | | | | |
| ILEXPARA_005969 | Putative membrane protein | | * | | | | | | | | | | | | |
| ILEXPARA_048009 | *panC*, pantothenate (vitamin B5) synthetase | | * | | | | | | | | | | | | |
| ILEXPARA_018064 | *arf*, auxin response factor | | | * | | | | | | | | | | | |
| ILEXPARA_019275 | Uncharacterized protein | | | * | | | | | | | | | | | |
| ILEXPARA_024153 | Uncharacterized protein | | | * | | | | | | | | | | | |
| ILEXPARA_035190 | Uncharacterized protein | | | * | | | | | | | | | | | |
| ILEXPARA_016483 | Hypothetical protein | | | | * | | | | | | | | | | |
| ILEXPARA_029421 | GCP4, gamma tubulin complex protein 4 | | | | | * | * | | | | | | | | |
| ILEXPARA_047849 | GOLS1, galactinol synthase 1 | | | | | * | * | | | | | | | | |
| ILEXPARA_003987 | NACK1, kinesin-like protein | | | | | * | * | | | | | | | | |
| ILEXPARA_044341 | Uncharacterized protein | | | | | * | | | | | | | | | |
| ILEXPARA_005359 | Uncharacterized protein | | | | | * | | | | | | | | | |
| ILEXPARA_035716 | RABE1C, ras-related protein | | | | | * | | | | | | | | | |
| ILEXPARA_010316 | MAPK, mitogen activated protein kinase | | | | | * | * | | | | | | | | |
| ILEXPARA_032923 | Uncharacterized protein | | | | | * | * | | | | | | | | |
| ILEXPARA_008149 | Uncharacterized protein | | | | | * | * | | | | | | | | |

*Appendix 1—table 3 continued on next page*

*Appendix 1—table 3 continued*

| Targets IDs | Description | miR159 | miR164 | miR167_1 | miR168 | miR169_2 | miR169_5 | miR171_1 | miR171_2 | miR390 | miR394 | miR396 | miR397 | miR398 | miR403 |
|---|---|---|---|---|---|---|---|---|---|---|---|---|---|---|---|
| ILEXPARA_048631 | Protein kinase | | | | | ✱ | ✱ | | | | | | | | |
| ILEXPARA_008152 | Uncharacterized protein | | | | | | | ✱ | | | | | | | |
| ILEXPARA_023090 | NAGK, N-acetyl-D-glucosamine kinase | | | | | | | ✱ | | | | | | | |
| ILEXPARA_024088 | RNA-binding (RRM/RBD/RNP motif) family protein | | | | | | ✱ | | | | | | | | |
| ILEXPARA_023716 | Endoglucanase | | | | | ✱ | | | | | | | | | |
| ILEXPARA_042182 | Uncharacterized protein | | | | | ✱ | | | | | | | | | |
| ILEXPARA_021515 | Pentatricopeptide repeat (PPR) protein | | | | | | | | ✱ | | | | | | |
| ILEXPARA_004925 | Uncharacterized protein | | | | | | | | | ✱ | | | | | |
| ILEXPARA_045111 | Rotamase FKBP 1 | | | | | | | | | ✱ | | | | | |
| ILEXPARA_013832 | ABCC2, ABC transporter C family member 2 protein | | | | | | | | | | ✱ | | | | |
| ILEXPARA_039828 | guaA, GMP synthase | | | | | | | | | | ✱ | | | | |
| ILEXPARA_028274 | Hypothetical protein | | | | | | | | | | ✱ | | | | |
| ILEXPARA_024538 | RPT6A, regulatory particle triple-A ATPase 6A | | | | | | | | | | | ✱ | | | |
| ILEXPARA_031387 | Uncharacterized protein | | | | | | | | | | | | | ✱ | |
| ILEXPARA_043757 | Uncharacterized protein | | | | | | | | | | | | | ✱ | |
| ILEXPARA_005297 | Uncharacterized protein | | | | | | | | | | | | | ✱ | |
| ILEXPARA_012032 | Uncharacterized protein | | | | | | | | | | | | | ✱ | |
| ILEXPARA_9682 | OST1B, oligosaccharyltransferase 1B | | | | | | | | | | | | | | ✱ |

## Appendix 1—key resources table

| Reagent type (species) or resource | Designation | Source or reference | Identifiers | Additional information |
|---|---|---|---|---|
| Gene (*Ilex paraguariensis*) | IpCS1 | GenBank | CAK9135737 | Xanthine methyltransferase gene of *Ilex paraguariensis* |
| Gene (*Ilex paraguariensis*) | IpCS2 | GenBank | CAK9135740 | 3-Methylxanthine methyltransferase gene of *Ilex paraguariensis* |
| Gene (*Ilex paraguariensis*) | IpCS3 | GenBank | CAK9135742 | Theobromine methyltransferase gene of *Ilex paraguariensis* |
| Strain, strain background (*Escherichia coli*) | BL21(DE3) | Novagen | 69450-M | Chemically competent cells |
| Biological sample (*Ilex paraguariensis*) | *Ilex paraguariensis* A. St.-Hil. var. *paraguariensis* | INTA-EEA Cerro Azul, Misiones, Argentina | cv CA 8/74 | Used to extract genomic DNA |
| Biological sample (*Ilex paraguariensis*) | *Ilex paraguariensis* A. St.-Hil. var. *paraguariensis* | Establecimiento Las Marías S.A.C.I.F.A., Corrientes, Argentina | cv SI-49 | Used to extract genomic DNA |
| Recombinant DNA reagent | pUC57-IpCS1 (plasmid) | GenScript | | Used to clone IpCS1 gene |
| Recombinant DNA reagent | pTrcHis-IpCS2 (plasmid) | This paper | | Used to clone IpCS2 gene |
| Recombinant DNA reagent | pUC57-IpCS3 (plasmid) | GenScript | | Used to clone IpCS3 gene |
| Recombinant DNA reagent | pUC57-AncIpCS1 (plasmid) | GenScript | | Used to clone AncIpCS1 gene |
| Recombinant DNA reagent | pUC57-AncIpCS2 (plasmid) | GenScript | | Used to clone AncIpCS2 gene |
| Sequence-based reagent | pET-15b- IpCS1 (plasmid) | This paper | | Used to express IpCS1 in *E. coli* BL21(DE3) |
| Sequence-based reagent | pET-15b- IpCS2 (plasmid) | This paper | | Used to express IpCS2 in *E. coli* BL21(DE3) |
| Sequence-based reagent | pET-15b- IpCS3 (plasmid) | This paper | | Used to express IpCS3 in *E. coli* BL21(DE3) |
| Sequence-based reagent | pET-15b- AncIpCS1 (plasmid) | This paper | | Used to express AncIpCS1 in *E. coli* BL21(DE3) |
| Sequence-based reagent | pET-15b- AncIpCS2 (plasmid) | This paper | | Used to express AncIpCS2 in *E. coli* BL21(DE3) |
| Sequence-based reagent | IpCS2F | This paper | PCR primers | 5'-ATGGACGTGAAGGAAGCAC-3' |
| Sequence-based reagent | IpCS2R | This paper | PCR primers | 5'-CTATCCCATGGTCCTGCTAAG-3' |
| Peptide, recombinant protein | IpCS1 | This paper | | Purified from *E. coli* BL21(DE3) cells |
| Peptide, recombinant protein | IpCS2 | This paper | | Purified from *E. coli* BL21(DE3) cells |
| Peptide, recombinant protein | IpCS3 | This paper | | Purified from *E. coli* BL21(DE3) cells |
| Peptide, recombinant protein | AncIpCS1 | This paper | | Purified from *E. coli* BL21(DE3) cells |
| Peptide, recombinant protein | AncIpCS2 | This paper | | Purified from *E. coli* BL21(DE3) cells |

*Appendix 1 Continued on next page*

*Appendix 1 Continued*

| Reagent type (species) or resource | Designation | Source or reference | Identifiers | Additional information |
|---|---|---|---|---|
| Commercial assay or kit | DNeasy Plant Mini Kit | QIAGEN | Cat. #: 69104 | Used to extract genomic DNA from *Ilex paraguariensis* |
| Commercial assay or kit | Quick-DNA HMW MagBead Kit | Zymo Research | Cat. #: D6060 | Used to extract genomic DNA from *Ilex paraguariensis* |
| Commercial assay or kit | Illumina TruSeq DNA Sample Preparation Kit | Illumina | Cat. #: FC-121-2003 | Used to construct paired-end libraries |
| Commercial assay or kit | Illumina Nextera Mate Pair Library Preparation Kit | Illumina | Cat. #: FC-132-1001 | Used to construct mate-pair libraries |
| Commercial assay or kit | Sequel Binding Kit 1.0 | Pacific Biosciences | Cat. #: 101-365-900 | Used for preparing DNA templates for sequencing on the PacBio Sequel System |
| Commercial assay or kit | Sequel Sequencing Kit 1.0 | Pacific Biosciences | Cat. #: 101-309-500 | Used to perform sequencing reactions on the PacBio Sequel System |
| Commercial assay or kit | SMRT Cell 1M | Pacific Biosciences | Cat. #: 100-171-800 | Consumable microchip used in the PacBio Sequel System for Single Molecule, Real-Time (SMRT) sequencing |
| Commercial assay or kit | pTrcHis TOPO TA Expression Kit | Invitrogen | Cat. #: K4410-01 | Used to clone IpCS2 gene |
| Commercial assay or kit | QIAEX II Gel Extraction Kit | QIAGEN | Cat. #: 20021 | Used to clone IpCS1, IpCS3, AnclpCS1, and AnclpCS2 genes into pET-15b expression vector |
| Commercial assay or kit | Agilent QuikChange Lightning Kit | Agilent Technologies Inc, Santa Clara, CA | Cat. #: 210518 | Used for site-directed mutagenesis of AnclpCS2 |
| Commercial assay or kit | QIAprep Spin Miniprep Kit | QIAGEN | Cat. #: 27104 | Used for the rapid purification of high-quality plasmid DNA |
| Commercial assay or kit | TALON spin columns | Takara Bio | Cat. #: 89068 | Used for the purification of histidine-tagged proteins |
| Chemical compound, drug | Xanthine | Sigma-Aldrich | Cat. #: X0626 | Used to test relative substrate preference of IpCS1–3 and AnclpCS1–2 |
| Chemical compound, drug | Xanthosine | Sigma-Aldrich | Cat. #: X0750 | Used to test relative substrate preference of IpCS1–3 and AnclpCS1–2 |
| Chemical compound, drug | 1-Methylxanthine | Sigma-Aldrich | Cat. #: 69720 | Used to test relative substrate preference of IpCS1–3 and AnclpCS1–2 |
| Chemical compound, drug | 3-Methylxanthine | Sigma-Aldrich | Cat. #: 222526 | Used to test relative substrate preference of IpCS1–3 and AnclpCS1–2 |
| chemical compound, drug | 7-Methylxanthine | Sigma-Aldrich | Cat. #: 69723 | Used to test relative substrate preference of IpCS1–3 and AnclpCS1–2 |
| Chemical compound, drug | Theobromine | Sigma-Aldrich | Cat. #: T4500 | Used to test relative substrate preference of IpCS1–3 and AnclpCS1–2 |
| Chemical compound, drug | Paraxanthine | Sigma-Aldrich | Cat. #: D5385 | Used to test relative substrate preference of IpCS1–3 and AnclpCS1–2 |
| Chemical compound, drug | Theophylline | Sigma-Aldrich | Cat. #: T1633 | Used to test relative substrate preference of IpCS1–3 and AnclpCS1–2 |
| Software, algorithm | Trimmomatic | DOI: 10.1093/bioinformatics/btu170 | v.0.39 | Used to remove adaptor contaminations and filter low-quality reads |
| Software, algorithm | Quake | DOI: 10.1186/gb-2010-11-11-r116 | v.0.3 | Used to correct clean reads |
| Software, algorithm | SOAPdenovo | DOI: 10.1186/2047-217X-1-18 | v.2 | Used to assemble and scaffold contigs |

*Appendix 1 Continued on next page*

*Appendix 1 Continued*

| Reagent type (species) or resource | Designation | Source or reference | Identifiers | Additional information |
|---|---|---|---|---|
| Software, algorithm | DeconSeq | DOI: 10.1371/journal.pone.0017288 | v.0.4.3 | Used to detect and remove sequence contaminants |
| Software, algorithm | Canu | DOI: 10.1101/gr.215087.116 | v.2.2 | Used for self-correction and assembly of long reads |
| Software, algorithm | PurgeHaplotigs | DOI: 10.1186/s12859-018-2441-2 | | Used to separate assembly haplotypes |
| Software, algorithm | Quickmerge | DOI: 10.1101/029306 | v.03 | Used to merge SOAPdenovo and Canu curated assemblies |
| Software, algorithm | SSPACE | DOI: 10.1093/bioinformatics/btq683 | v.2.1.1 | Used to refine scaffolds and contigs |
| Software, algorithm | RepeatMasker | http://repeatmasker.org/ | | Used to mask the genome assembly |
| Software, algorithm | Funannotate | DOI: 10.5281/zenodo.2604804 | v.1.8.13 | Used to predict the protein- and non-coding genes |
| Software, algorithm | Infernal | DOI: 10.1093/bioinformatics/btt509 | v.1.1.4 | Used to improve the prediction of small RNAs and microRNAs |
| Software, algorithm | tRNAScan-SE | DOI: 10.1007/978-1-4939-9173-0_1 | v.2.0 | Used to improve the prediction of transfer RNAs |
| software, algorithm | TAPIR | http://bioinformatics.psb.ugent.be/webtools/tapir | | Used to identify miRNA targets |
| Software, algorithm | TargetFinder | DOI: 10.1007/978-1-60327-005-2_4 | v.1.7 | Used to identify miRNA targets |
| Software, algorithm | InterProScan | DOI: 10.1093/bioinformatics/btu031 | v.5.55-88.0 | Used to assign function of the predicted genes |
| Software, algorithm | eggNOG-mapper | DOI: 10.1093/nar/gky1085 | v.2.1.7 | Used to assign function to the predicted genes |
| Software, algorithm | Dfam TE Tools | https://github.com/Dfam-consortium/TETools | v.1.5 | Used to estimate the repeat content |
| Software, algorithm | CoGe's tool SynMap | https://genomevolution.org/ | | Used to estimate rates of synonymous substitution (Ks) between paralogous and orthologous genes |
| Software, algorithm | CoGe's tool SynFind | https://genomevolution.org/ | | Used to determine the syntenic depth ratio between *I. paraguariensis*, *C. canephora*, and *V. vinifera* |
| Software, algorithm | CoGe's tool GEvo | https://genomevolution.org/ | | Used to compare CS and XMT syntenic regions |
| Software, algorithm | MAFFT | DOI: 10.1093/molbev/mst010 | v.7.0 | Used to align amino acid sequences |
| Software, algorithm | FastTree | DOI: 10.1371/journal.pone.0009490 | v.2 | Used to perform phylogenetic analysis of SABATH sequences |
| Software, algorithm | IQTree | DOI: 10.1093/nar/gkw256 | | Used to estimate ancestral sequences |
| Software, algorithm | Phenix | DOI: 10.1107/S0907444909052925 | | Used to solve the crystal structure of IpCS3 |
| software, algorithm | REFMAC5 | DOI: 10.1107/S0907444911001314 | | Used to refine the crystal structure of IpCS3 |
| Software, algorithm | COOT | DOI: 10.1107/S0907444910007493 | v.0.9.8.3 | Used to refine the crystal structure of IpCS3 |
| Other | *Ilex paraguariensis* transcriptome sequence data | ENA | PRJNA315513 | Used to assess the completeness of *Ilex paraguariensis* genome |
| Other | *Ilex paraguariensis* transcriptome sequence data | NCBI | SRP043293 | Used to assess the completeness of *Ilex paraguariensis* genome |

*Appendix 1 Continued on next page*

*Appendix 1 Continued*

| Reagent type (species) or resource | Designation | Source or reference | Identifiers | Additional information |
|---|---|---|---|---|
| Other | *Ilex paraguariensis* transcriptome sequence data | NCBI | SRP110129 | Used to determine the expression of IpCS1–5 genes |
| Other | Vivaspin columns | Sartorius | Cat. #: VS0101 | Used to remove proteins after enzymatic reaction |
| Other | Kinetex 5 µM EVO C18 column | Phenomenex | Cat. #: 00F-4467-AN | Used for high-performance liquid chromatography |
| Other | Crystal Gryphon robot | Art Robbins Instruments | Cat. #: 100-1010 | Used for automating crystallization |

