## [Editor Report]

This very important study combines genomics, biochemistry, structural biology and ancestral sequence reconstruction to address the basis of caffeine biosynthesis in Yerba mate, a species that is phylogenetically unrelated to other plants, namely coffee and tea, in which this pathway has been studied before. The manuscript reports the first draft genome sequence for yerba mate and provides convincing evidence for the identity and characteristics of enzymes for caffeine biosynthesis. The authors are able to propose structural constraints for the convergent evolution of individual reactions. The work will be of interest to plant and evolutionary biologists and anyone studying natural product biosynthesis.

---

## [Decision Letter]

[Editors' note: this paper was reviewed by Review Commons.]

---

## [Author Response]

*General Statements*

We are grateful to the reviewers for the time they invested in reviewing our manuscript and for their supportive feedback, which acknowledges our study as a distinguished, well-written comprehensive contribution to the field of plant biology.

We also thank them for recognizing our study as appealing to a wide audience. Our work focuses on a plant that impacts the daily lives of millions of people and generates considerable income around much of the world. They also mention that our work is integrative across several fields of inquiry, from evolutionary to structural biology, and as such should attract scientific attention from many diverse researchers.

We have carefully addressed all the minor corrections and text modifications suggested by the reviewers, which have notably improved the overall quality and readability of our manuscript. Based on the feedback provided by the reviewers, we have conducted site-directed mutagenesis studies and comparative bioinformatic analyses, which enhanced our understanding of the structural basis underlying the substrate selectivity of CS/XMT enzymes in yerba mate and other caffeine-producing plants (coffee, tea, cocoa, orange, and guaraná). Site-directed mutagenesis studies were performed by Emily M. Catania, resulting in her inclusion as a co-author on this work. Her authorship and placement in the author list have been approved by all co-authors. Additionally, we made adjustments to comply with *eLife’*s requirements. Specifically, we have added a Key Resources Table as an appendix section, and incorporated the Supplementary Material into the main manuscript as an appendix section, figure supplements, tables, and source data. A detailed point-by-point response to each of the reviewers' comments is provided below. Changes made to the revised manuscript are highlighted in red.

Reviewer #1:"Yerba mate (Ilex paraguariensis) genome provides new insights into convergent evolution of caffeine biosynthesis" by Vignale et al., reported the first draft genome of Yerba mate (Ilex paraguariensis) and explored the secret for the convergent evolution of caffeine biosynthesis in this famous caffeine-producing plant.The authors found that Yerba mate caffeine biosynthesis enzyme was converged upon a different biochemical pathway compared to that of coffee and tea.The crystal structure for the caffeine synthase enzyme from the Yerba plant showed that the convergent solutions have evolved for substrate positioning because different amino acid residues facilitate a different substrate orientation such that efficient methylation occurs in the independently evolved enzymes in Yerba mate and coffee.So the authors claimed that although they did not find evidence for convergent genes opted for caffeine biosynthesis by using yerba mate genome sequence, their x-ray diffraction data on caffeine synthase IpCS3 showed that convergent three dimensional enzyme protein structures can confer convergent substrate interactions and catalysis by the enzymes. They showed that structural constraints are minimal for the convergent evolution of individual reactions.So, this manuscript not only reported the first genome sequence for Yerba mate, and uncovered the genetic, biochemical and structural bases for convergent evolution of caffeine in yerba mate, but also extend the previous concept of convergent evolution by gene co-option to the three dimensional structural option to the convergent substrate interactions in enzymatic catalyses, despite of the weak support.Different from these in XMT-type enzymes in Coffea and Citrus, but similar to CS-type enzymes in Camellia, Theobroma and Paullinia, three duplicated IpCS1, IpCS2, and IpCS3, methylates X to form 3X, methylates 3X to produce TB, and methylates TB to form CF, respectively. The study also dissected the genomic basis for the possible selection and evolution of these genes from SABATH enzyme family and assembly the complete biosynthesis pathway for caffeine in yerba mate. e.g. lost of XMT orthologues responsible for caffeine biosynthesis in Coffea and Citrus, but evolve tandem duplicated CSs in yerba mate (I. paraguariensis), and only in Ip, but not in related species I. polyneura and I. latifolia.The manuscript was well written and presented, with clear descriptions on the research backgrounds and existing questions remain to be answered. The logic of the manuscript presentation is also clear.”

We are very pleased that Reviewer #1 recognized the importance of the multiple integrated lines of evidence we provided to unveil the convergent evolution of caffeine biosynthesis in Yerba mate.

Reviewer #1:“Weak points:1. I do not see the vigorous investigation of structural characters of IpCS1,2,3, based on x-ray structural data of IpCS3, only slightly touching on the docking models of xanthine alkaloids in IpCS1 and IpCS2. no mutation of these well-known key amino acid residues in CS enzymes of coffea and tea plants.”

We had not intended on pursuing mutagenesis to investigate which amino acids contributed to the evolution of caffeine biosynthesis. However, in response to this constructive feedback of reviewer #1, we performed site-directed mutagenesis of the ancient enzyme, AncIpCS2 which gave rise to IpCS3, responsible for the final step of caffeine formation in YM. The mutant enzymatic data are now shown in Figure 5 and clearly show that the evolution of TB methylation preference to produce caffeine depends upon the mutated sites that have been shown to be important in caffeine synthases from other species like tea, guarana, and cacao. This fascinating result strengthens our manuscript and further extends the levels of convergent evolution of caffeine biosynthesis to the level of mutated amino acid positions in the enzymes.

2. “Comparisons of the biochemical properties of IpCS1,2,3, only enzymatic reactions and substrate preference tests, but did not really compare these with their enzyme amino acid sequences with these from other plants, such as tea and coffea. The really special characteristics of these IpCS1,2,3 enzymes need enzyme sequence comparison, key amino acid mutation, compared with these characterized CS enzymes.”

In response to this useful comment and to deepen our understanding of substrate selectivity in CS/XMT enzymes, we compared the CS enzymes in yerba mate with those of other caffeine-producing plants (coffee, tea, cocoa, orange, and guaraná) through a multiple amino acid alignment. Although our initial Supplementary Figure 10 showed partial alignments, it did not include tea or coffee. The new complete multiple alignment, depicted in Figure 6—figure supplements 3 and 4, shows convergent changes predicted to participate in substrate binding and promote methylation preference switches. Furthermore, our mutagenesis studies experimentally demonstrate the functional importance of some of the homologous amino acid positions shown as stated above.

3. “Although they are phylogenetically unrelated plants, yerba mate, tea and coffea plants have similar CS enzymes and characters of metabolic pathways, and should make comparisons.”

We agree with reviewer #1’s suggestion, and for that reason, we have conducted further comparative analyses based on the amino acid alignment of CS/XMT enzymes (Figure 6—figure supplements 3 and 4) and site-directed mutagenesis of AncIpCS2 followed by enzyme activity assays (Figure 5).

Reviewer #1:“Although it is clear that caffeine biosynthesis in these phylogenetically unrelated plant species are convergently evolved through independent recruitment and co-option of SABATH family enzymes, there are at least five different pathways towards caffeine biosynthesis, with substrate diversity and enzyme sequence diversity in these caffeine-producing plants. Therefore, the convergent evolution features in other caffeine-producing plants remain to be unexplored.The manuscript reported the first draft genome sequence for yerba mate, identified the genetic secret for co-opted genes IpCS1,2,3, but less import IpCS4,5, for caffeine biosynthesis. The authors carefully examined their enzyme activity towards various substrates to unveil their substrate preference and probe the possible evolution routine.Furthermore, by obtaining the x-ray crystal structure of IpCS3 and docking model analysis of IpCS1,2 for enzyme-substrate complexes , the authors put forward a structural constraints being the minimal for the convergent evolution of individual reactions.These data advanced our understanding of diverse features of the convergent caffeine biosynthesis evolution in these phylogenetically unrelated plants. As a tea plant scientist, the study provides a nice reference for us to understand these wild tea species that do not produce caffeine.”

The insightful comments of Reviewer #1 have prompted our manuscript revision such that it is now much improved with more clear connections to what is known from tea, in particular, due to our new bioinformatic and experimental mutagenesis studies. As a result, we are confident that the findings have advanced our understanding of plant evolutionary biology.

Reviewer #2:“The entire study is methodically crafted and can be largely considered a comprehensive effort. It integrates genomics, biochemistry, and structural biology to tackle the complex question concerning the evolution of the caffeine metabolic pathway in herba mate. The findings of this work and the narrative hold considerable appeal for a wide audience, appealing both the scientific (plant biology, evolutionary biology) and lay communities, given the widespread consumption of beverages like coffee, tea, and mate.The current manuscript is destined for publication in a high-profile journal, not only for its exemplar quality but also because it serves as a cornerstone for fostering genomic and bioinformatic capabilities within the South American scientific landscape. Its approach stands as a quintessence work, distinguished among a flood of plant genome publications marred by irrelevant and redundant analyses.While the absence of genomes from other plant species within the same family could have enriched the analysis, such considerations are understandably beyond the scope of the current manuscript.”

We are very pleased to know that the high impact of our integrative research approach and findings are evident.

“My only concern, however, pertains to the significance of the findings. As noted by the authors, examples of convergent evolution in caffeine biosynthesis have been well-documented elsewhere. My only suggestion to overcome this weakness is to include a genomics comparison between the different evolution events regarding caffeine biosynthesis at different lineages pointing towards when (time) it happens.”

This reviewer has raised an intriguing point that we had contemplated in this work and our previously published studies but have avoided addressing due to the inherent uncertainty associated with fine-scale molecular dating analyses. Nonetheless, it is interesting to note that, at least for Yerba mate, Tea, Citrus, Guarana and Coffee, they all appear to have diverged from their non-caffeine-producing relatives within the last 10-20 million years or less. In other words, the convergence is relatively recent and broadly concomitant for all known lineages except the Malvales. There it remains unclear whether there is a single ancient origin of caffeine biosynthesis or multiple more recent ones in the Kola nut, Chocolate, and Basswood flower.

“It is rather imperative to rectify the terminology concerning the taxonomic relationships among coffee, tea, and yerba mate. While colloquially referred to as "relatives," their evolutionary lineages diverge significantly over l time scale. Hence, it would be more accurate to restrict the term "relatives" to species within the same family.”

We have taken into account the comment from reviewer #2 and corrected the text to no longer refer to coffee and tea as "relatives" of yerba mate.

Reviewer #2:“The current work combines genomics, biochemistry, structural biology and ancestral sequence reconstruction to address the evolution and functionality of the methyltransferases act in caffeine biosynthesis in Yerba mate.The data are clear. Though the findings will benefit by crystal structures of the three methyltranferases.”

Although we agree with reviewer #2 that the work would benefit from crystal structures of all three enzyme complexes rather than just IpCS3, which we did thoroughly characterize; however, obtaining the remaining structures for IpCS1 and IpCS2 is a challenging and time-consuming task beyond the scope of this study. Nevertheless, to address this point, we have created AlphaFold2 models of all three enzymes (IpCS1, IpCS2, and IpCS3) and conducted molecular docking simulations with their substrates, which are depicted in Figure 7 and Figure 7—figure supplement 1. To acknowledge the point of the reviewer, we now state that if crystal structures could be generated in the future, our understanding of the convergent activities would be enhanced. Of course, not all proteins crystallize well, with or without ligands bound, so it remains to be seen if these experiments will be fruitful.

“More explanations and details regarding ancestral reconstruction might be necessary.”

We have addressed reviewer #2's feedback and included additional information on the methods used in our ancestral reconstruction analysis in the revised manuscript.

“Additionally I will recommend a bayesean based phylogeny for the NMT enzymes to have a better picture for the evolution timeframe.”

Unfortunately, it is not possible to date the origins of NMT enzymes directly from a gene tree due to the fact that there are numerous gene duplication events. To be able to pinpoint whether duplication events occurred prior to or after species divergences would require much more NMT sequence data than is available for the closest relatives of caffeine-producing plants. Such fine-scale phylogenomic analyses would require a massive sequencing approach which is outside of the scope of this study. Instead, it is possible to assess the species tree divergence times for each lineage of caffeine-producing species to place bounds on the timing of origins of caffeine biosynthesis. Therefore, we have now scrutinized previously published studies that provide divergence time estimates for the caffeine-producing lineages. We have reported these published divergence times in the text to address this interesting point.

Reviewer #3:“Vignale et al. report on the genome assembly of Yerba mate tea, which is used to make the third most consumed caffeinated beverage. Through evolutionary analysis of the genome, the authors confirmed the convergent evolution of caffeine biosynthesis in Yerba Mate and then focused on the three N-methyltransferases involved in caffeine synthesis. The authors solved the crystal structure of IpCS3, which methylates TB to produce caffeine. The manuscript is a good and comfortable read, well illustrated, and the conclusions are supported by compelling data.”

We are pleased that the diverse data presented have resulted in a clear and integrative set of results that clarify the complexity of the convergent evolution of caffeine biosynthesis in YM.

Reviewer #3:“There are a few minor issues here that need to be addressed:1. The authors only compared the active site of IpCS3 with that in coffee. There should be a comprehensive comparison of the substrate binding sites of N-methyltransferases in coffee, tea, and yerba mate, which could provide structural evidence for the convergent evolution of N-methyltransferases, as whas was done in Nat. Commun, 2020, 11:1473.”

We thank reviewer #3 for the insightful suggestion, which aligns with the feedback provided by reviewer #1. To provide a comprehensive comparison of the substrate binding sites of N-methyltransferases in yerba mate and other caffeine-producing plants (coffee, tea, cocoa, orange, and guaraná), we performed a multiple alignment of their amino acid sequences (Figure 6—figure supplements 3 and 4). This alignment shows convergent changes predicted to participate in substrate binding and promote methylation preference switches. We have also compared the crystal structure for IpCS3 to that of Theacrine synthase from *Camellia assamica*, the most similar crystal structure that was previously reported in Nat. Commun, 2020, 11:1473. This comparison shows important amino acid residues within the active sites of these enzymes that appear to be involved in substrate interactions. And, most importantly, we have provided experimental verification of the functional importance of those sites for the evolution of TB methylation preference to produce caffeine.

2. The wavelengths used for one HPLC run are ONLY mentioned in the Methods section. Please add the wavelengths used in Figure 4b, Supplementary Figure 6 and the corresponding legend.”

In order to clarify this detail, we have revised the legend for Figure 4B and that of Figure 5—figure supplement 5 (previously Supplementary Figure 6) to state that absorbance at 254 nm is shown.

3. Theobromine (TB) is not present in Supplementary Figure 7. Please correct the legend”.

We have addressed reviewer #3's comment and corrected the legend of Figure 6—figure supplement 1 (previously Supplementary Figure 7).

4. Kinetic studies were performed. The authors have provided only one table (Supplementary Table 7) to show the results. It would be better to show all curves.”

As suggested, we have included all kinetic curves in Figure 4—figure supplement 1.

Reviewer #3:“This work will contribute to the study of plant evolution and the field of plant secondary metabolites, especially xanthine alkaloids.”

Reply to reviewer #3: The numerous suggestions made by Reviewer #3 have led to revisions that improved the manuscript and will ensure that the results we have reported have a high impact and contribute to the field of plant-specialized metabolite biochemistry and evolution.